# In vivo recording of suprachiasmatic nucleus dynamics reveals a dominant role of arginine vasopressin neurons in circadian pacesetting

**Yusuke Tsuno[1], Yubo Peng[1], Shin-ichi Horike[2], Mohan Wang[1], Ayako Matsui[1], Kanato Yamagata[3], Mizuki Sugiyama[4], Takahiro J. Nakamura[4], Takiko Daikoku[5], Takashi Maejima[1], Michihiro Mieda[1]***

1 Department of Integrative Neurophysiology, Graduate School of Medical Sciences, Kanazawa University, Kanazawa, Japan, 2 Division of Integrated Omics Research, Research Center for Experimental Modeling of Human Disease, Kanazawa University, Kanazawa, Japan, 3 Child Brain Project, Tokyo Metropolitan Institute of Medical Science, Tokyo, Japan, 4 Laboratory of Animal Physiology, School of Agriculture, Meiji University, Kawasaki, Japan, 5 Division of Animal Disease Model, Research Center for Experimental Modeling of Human Disease, Kanazawa University, Kanazawa, Japan

* mieda@med.kanazawa-u.ac.jp

**Data Availability Statement:** All relevant data are within the paper and its Supporting Information files.

## Abstract

The central circadian clock of the suprachiasmatic nucleus (SCN) is a network consisting of various types of neurons and glial cells. Individual cells have the autonomous molecular machinery of a cellular clock, but their intrinsic periods vary considerably. Here, we show that arginine vasopressin (AVP) neurons set the ensemble period of the SCN network in vivo to control the circadian behavior rhythm. Artificial lengthening of cellular periods by deleting *casein kinase 1 delta* (*CK1δ*) in the whole SCN lengthened the free-running period of behavior rhythm to an extent similar to *CK1δ* deletion specific to AVP neurons. However, in SCN slices, PER2::LUC reporter rhythms of these mice only partially and transiently recapitulated the period lengthening, showing a dissociation between the SCN shell and core with a period instability in the shell. In contrast, in vivo calcium rhythms of both AVP and vasoactive intestinal peptide (VIP) neurons in the SCN of freely moving mice demonstrated stably lengthened periods similar to the behavioral rhythm upon AVP neuron-specific *CK1δ* deletion, without changing the phase relationships between each other. Furthermore, optogenetic activation of AVP neurons acutely induced calcium increase in VIP neurons in vivo. These results indicate that AVP neurons regulate other SCN neurons, such as VIP neurons, in vivo and thus act as a primary determinant of the SCN ensemble period.

## Introduction

In mammals, the suprachiasmatic nucleus (SCN) of the hypothalamus functions as the central clock to control multiple circadian biological rhythms of behaviors and physiological functions, such as sleep-wakefulness, body temperature, and hormone secretion [1]. Most of approximately 20,000 SCN neurons are GABAergic and include several neuron types characterized by coexpressing peptides. For example, vasoactive intestinal peptide (VIP)-positive

**Funding:** This work was supported in part by JSPS KAKENHI Grant Numbers JP20K07259, JP23K06345 (to Y.T.); JP22K20738 (to A.M.); JP18H04972, JP18K19421, JP20K21498, JP22H02802 (to M.M.) (Japan Society for the Promotion of Science: https://www.jsps.go.jp/english/index.html); JST SPRING Grant Number JPMJSP2135 (to Y.P., M.W.) (Japan Science and Technology Agency: https://www.jst.go.jp/EN/); the Takeda Science Foundation (to M.M.) (https://www.takeda-sci.or.jp/en/); the Naito Foundation (to M.M.) (https://www.naito-f.or.jp/en/); the Japan Foundation for Applied Enzymology (to M.M.) (https://www.jfae.or.jp/); and Kanazawa University CHOZEN project (to M.M.) (http://www.o-fsi.kanazawa-u.ac.jp/research/chozen/). The funders had no role in study design, data collection and analysis, decision to publish, or preparation of the manuscript.

**Competing interests:** The authors have declared that no competing interests exist.

**Abbreviations:** AVP, arginine vasopressin; CK1δ, casein kinase 1 delta; CRE, cAMP responsive element; CT, circadian time; DRD1a, D1a dopamine receptor; LD, light–dark; MEF, mouse embryonic fibroblast; NMS, neuromedin S; PFA, paraformaldehyde; ROI, region of interest; SCN, suprachiasmatic nucleus; tTA, tetracycline transactivator; TTFL, transcription–translation feedback loop; VIP, vasoactive intestinal peptide.

neurons in the ventral core region and arginine vasopressin (AVP)-positive neurons in the dorsal shell region are 2 representative neuron types in the SCN [1]. In individual cells, the molecular machinery of cellular clocks is driven by the autoregulatory transcription–translation feedback loop (TTFL), in which transcriptional activators CLOCK and BMAL1 play a central role [2]. *Period* (*Per1*, *2*, and *3*) and *Cryptochrome* (*Cry1* and *2*) are 2 of many target genes of CLOCK/BMAL1. PER and CRY proteins then repress CLOCK/BMAL1 activity, completing a negative feedback loop. Casein kinase 1 delta (CK1δ) has been known as a critical regulator of the period length of cellular clocks. By phosphorylating PER2 protein, CK1δ regulates the speed of degradation and nuclear retention of PER2 and, thereby, other clock proteins [3–6]. Indeed, pharmacological and genetic experiments revealed that eliminating CK1δ activity lengthened periods of circadian behavior rhythm and molecular oscillations in the SCN and peripheral cells [4,7,8]. Intriguingly, these intracellular molecular mechanisms are not unique to SCN neurons but are common to peripheral cells. Instead, intercellular communications among SCN cells are likely essential for the SCN to generate a highly robust, coherent circadian rhythm as the central clock [1].

Thus, SCN contains tens of thousands of cellular clocks in various types of neurons and glial cells. However, these cell-autonomous rhythms are sloppy, with a considerable variation in the period length [1,9–13]. These facts raise a fundamental question on how the ensemble amplitude, period, and phase of the circadian rhythm at the SCN network level are determined. VIP has been known as the most critical contributor to the synchronization among SCN neurons and is also involved in the photoentrainment to regulate the ensemble phase of the SCN according to the light/dark cycle [14–20]. On the other hand, recent studies utilizing cell type–specific genetic manipulations of the TTFL in mice implicated that neuromedin S (NMS)-, AVP-, D1a dopamine receptor (DRD1a)-, and VIP receptor VPAC2-expressing neurons and even astrocytes affect pacemaking of the SCN network [21–27]. NMS neurons were reported to include AVP, VIP, and other types of neurons and act as pacemakers essential for the generation and period-setting of the SCN ensemble rhythm measured by both wheel-running behavior (in vivo) and PER2::LUC reporter expression in slices (ex vivo) [21]. In contrast, Patton and colleagues showed that neurons labeled by a *Vpac2-Cre* driver line are primarily distributed in the SCN shell and include most AVP neurons and few VIP neurons [28]. They regulate the period and coherence of circadian wheel-running rhythm but also require the cooperative action of VIP neurons for pacemaking PER2::LUC rhythms in SCN slices [25,28].

AVP neuron-specific disruption of cellular clocks by deleting *Bmal1* disturbed substantially, but not completely abolished, the circadian rhythm of locomotor activity (home-cage activity) [22]. Also, manipulating the cellular circadian period of AVP neurons by deleting or overexpressing *CK1δ* lengthened or shortened the free-running period of locomotor activity rhythm accordingly [23]. These observations support a significant role for AVP neurons in the period-setting and coherence of the SCN ensemble rhythm. However, the behavioral period lengthening by the specific deletion of *CK1δ* was not fully recapitulated in cellular PER2::LUC rhythms in SCN slices [23], implicating different states of the SCN network between in vivo and ex vivo. In addition, the extent to which AVP neurons contribute to the SCN pacemaking relative to other types of SCN neurons remains unclear. To address these questions, we compared the period-lengthening effects caused by *CK1δ* deletion in the entire SCN to AVP neuron-specific deletion in both behavior and slice rhythms. In addition, we examined whether the behavioral period lengthening due to AVP neuron-specific *CK1δ* deletion is recapitulated in the SCN cellular clock oscillations in vivo by monitoring the intracellular $Ca^{2+}$ ($[Ca^{2+}]_i$) rhythms of AVP and VIP neurons using fiber photometry in freely behaving animals. Our results suggested that AVP neurons play a principal role in the circadian pacesetting of the SCN network in vivo.

## Results

### Deletion of *CK1δ* in the entire SCN lengthens the circadian period of behavior comparably to AVP neuron-specific *CK1δ* deletion

We previously showed that lengthening the cellular circadian period of AVP neurons by the specific deletion of *CK1δ* lengthened the free-running period of circadian behavior, indicating that AVP neurons are involved in setting the ensemble period of the SCN network [23]. The remaining question was how much AVP neurons contribute to the period-setting. To elucidate this, we aimed to compare the period lengthening caused by AVP neuron-specific *CK1δ* deletion to that by pan-SCN *CK1δ* deletion. Although CaMKIIα is not abundant in the SCN [29], a particular *CaMKIIα-Cre* line [30] drives Cre expression in the forebrain, including the entire SCN [31]. Indeed, crossing this line to floxed *Bmal1* mice resulted in >90% deletion of BMAL1 in the SCN and a complete loss of circadian behavior rhythm [31]. We also reproduced the arrhythmicity of *CaMKIIα-Cre; Bmal1$^{flox/flox}$* mice, confirming *CaMKIIα-Cre* mice as a pan-SCN Cre driver (S1A and S1B Fig).

By crossing this *CaMKIIα-Cre* line with *CK1δ$^{flox/flox}$* [8], we deleted *CK1δ* in the forebrain and the entire SCN neurons (*CaMKIIα-CK1δ$^{-/-}$*). *CaMKIIα-CK1δ$^{-/-}$* mice showed normal diurnal locomotor activity (home-cage activity) rhythm in the 12 h of light and 12 h of darkness (LD) condition (Fig 1A and 1B). In constant darkness (DD), *CaMKIIα-CK1δ$^{-/-}$* mice showed a longer free-running period (24.83 ± 0.08 h) than that of control mice (23.93 ± 0.03 h, $P < 0.001$, Fig 1C). By contrast, their amplitude of locomotor activity rhythm (Qp) was normal ($P = 0.52$, Fig 1D). The difference in the free-running period between control and *CaMKIIα-CK1δ$^{-/-}$* mice was almost similar to the difference between control and *Avp-CK1δ$^{-/-}$* mice (23.94 ± 0.03 h versus 24.72 ± 0.03 h) [23], as well as with that between control and *Vgat-Cre; CK1δ$^{flox/flox}$* mice (approximately 40-min elongation), another line with pan-SCN neuronal *CK1δ* deficiency [32]. These data suggest that AVP neurons are the principal determinant of the free-running period of the circadian behavior rhythm.

To examine the contribution of VIP neurons in the setting of the SCN ensemble period, we deleted *CK1δ* specifically in VIP neurons by crossing *Vip-ires-Cre* mice [33] with *CK1δ$^{flox/flox}$* (*Vip-CK1δ$^{-/-}$*). However, we did not observe any change in locomotor activity rhythm of *Vip-CK1δ$^{-/-}$* mice (Fig 2A–2D, period, control 23.93 ± 0.04 versus *Vip-CK1δ$^{-/-}$* 23.91 ± 0.03). This result was consistent with a previous report that the lengthening of the intrinsic TTFL period in VIP neurons by the overexpression of *Clock$^{Δ19}$* did not alter the behavioral free-running period [21]. Thus, although VIP peptide act as an essential synchronizer of SCN neurons, the TTFL in VIP neurons itself may have little contribution to the ensemble period-setting of the SCN network.

### Cellular clocks in AVP neurons are necessary for the sustained generation of the circadian wheel-running rhythm

We previously reported that disruption of cellular clocks in AVP neurons by deleting *Bmal1* (*Avp-Bmal1$^{-/-}$* mice) causes severe attenuation of circadian behavior rhythm measured by home-cage activity [22]. However, most *Avp-Bmal1$^{-/-}$* mice still retained some circadian rhythmicity with unstable, lengthened free-running period and activity time. In contrast, Lee and colleagues reported that mice with *Bmal1* deletion in NMS neurons (*Nms-Bmal1$^{-/-}$* mice) lost the circadian rhythm in wheel-running behavior [21]. Notably, a recent study showed that rhythms of home-cage activity and body temperature in *Avp-Bmal1$^{-/-}$* mice were very similar to those in *Nms-Bmal1$^{-/-}$* mice [34]. Thus, circadian phenotypes may be significantly different between wheel-running and home-cage activity rhythms.

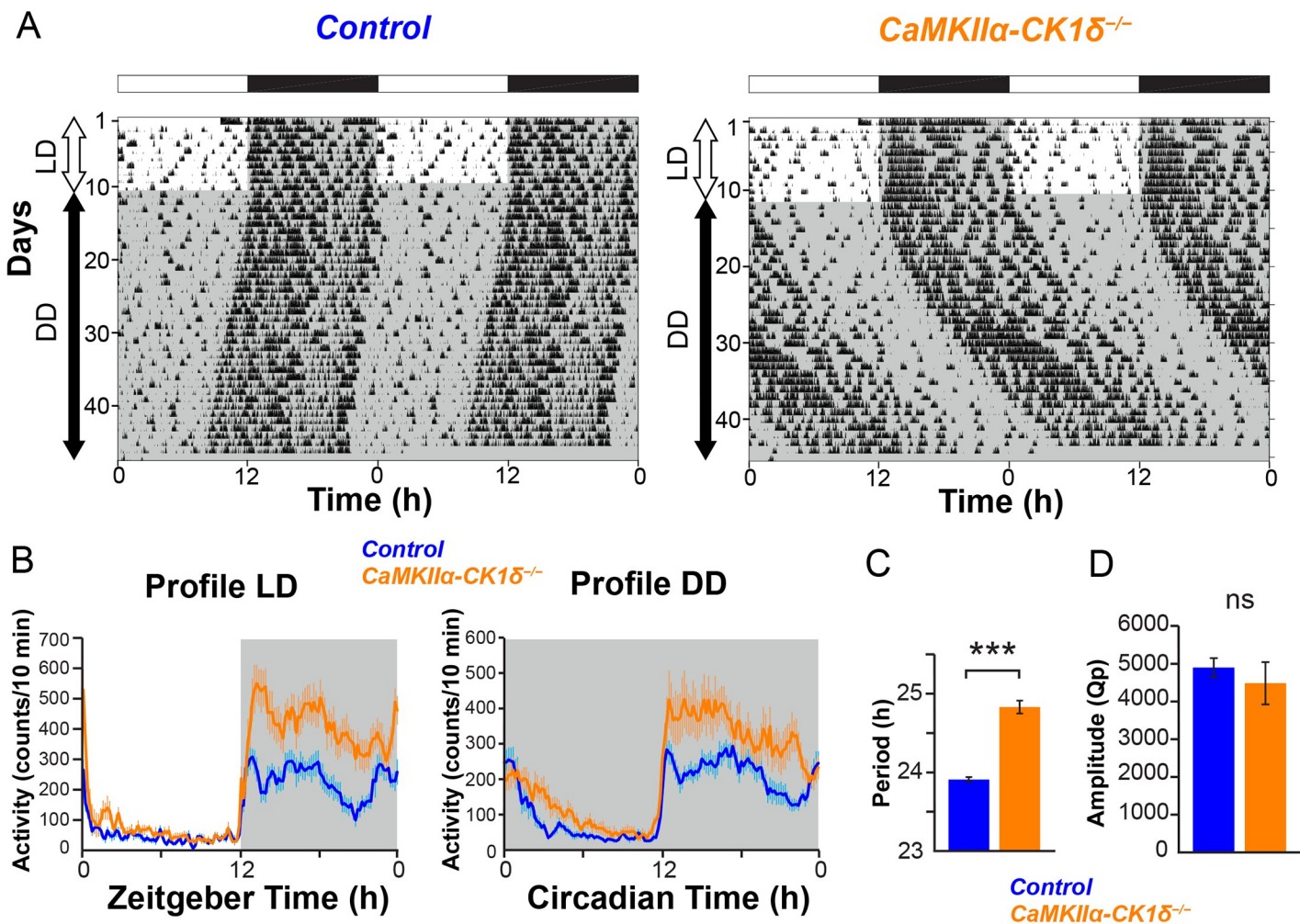

**Fig 1. CaMKIIα-CK1δ⁻/⁻ mice show lengthening of the free-running period in DD.** (**A**) Representative locomotor activity of control and *CaMKIIα-CK1δ⁻/⁻* mice (home-cage activity). Animals were initially housed in 12:12-h LD conditions and then transferred to DD. Gray shading indicates the time when lights were off. (**B**) Averaged daily profile of locomotor activity in LD (left) or DD (right). (**C**) The free-running period in DD. (**D**) The circadian amplitude of locomotor activity rhythms (Qp values obtained from periodogram analyses). Values are mean ± SEM; *n* = 10 for control, *n* = 13 for *CaMKIIα-CK1δ⁻/⁻* mice. The underlying data can be found in S1 Data. ***$P < 0.001$ by two-tailed Student *t* tests; ns, not significant. See also S1 and S2 Figs.

Given the critical role of SCN AVP neurons in setting the free-running period, we reevaluated the circadian behavior rhythm of *Avp-Bmal1⁻/⁻* and *Avp-CK1δ⁻/⁻* mice by measuring wheel-running activity. The difference in wheel-running rhythm between *Avp-CK1δ⁻/⁻* and control mice was consistent with that of home-cage activity rhythm [23], namely, approximately 50-min lengthening of the free-running period (control 23.73 ± 0.05 versus *Avp-CK1δ⁻/⁻* 24.52 ± 0.07, $P < 0.001$, S2A and S2B Fig). *Avp-Bmal1⁻/⁻* mice also demonstrated an impaired wheel-running rhythm similar to the previously described home-cage activity rhythm, i.e., "split" behavior with a lengthened free-running period in DD. However, in contrast to the home-cage activity, 7 out of 8 *Avp-Bmal1⁻/⁻* mice gradually lost the circadian wheel-running rhythm after approximately 10 to 20 days in DD (S2C Fig). Such a gradual disappearance of the wheel-running rhythm was similar to that of *Nms-Bmal1⁻/⁻* mice [21]. Therefore, the impairment of circadian behavior rhythm may be comparable between *Avp-Bmal1⁻/⁻* and *Nms-Bmal1⁻/⁻* mice.

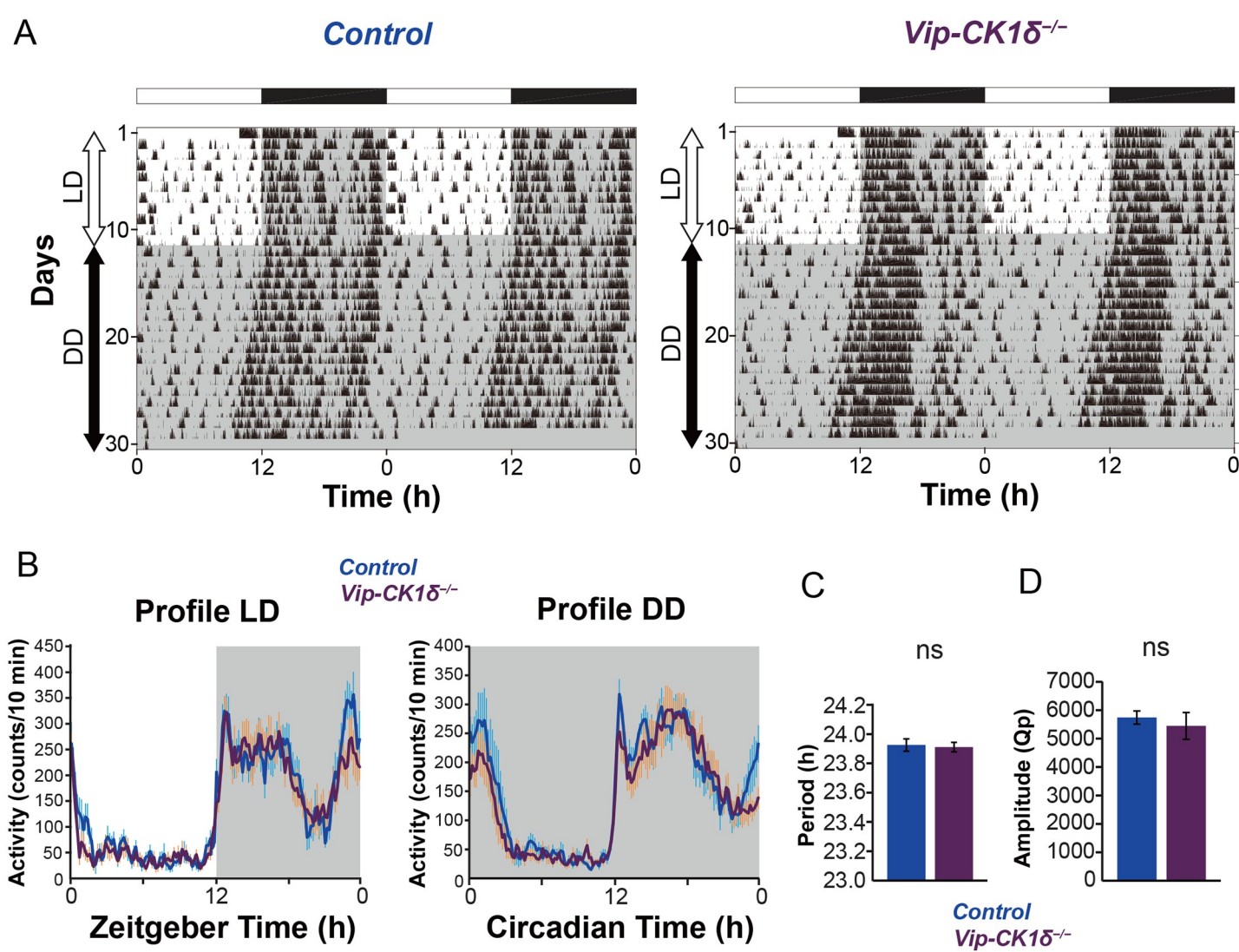

**Fig 2. *Vip-CK1δ⁻/⁻* mice show no change in the free-running period in DD.** (**A**) Representative locomotor activity of control and *Vip-CK1δ⁻/⁻* mice (home-cage activity). Gray shading indicates the time when lights were off. (**B**) Averaged daily profile of locomotor activity in LD (left) or DD (right). (**C**) The free-running period in DD. (**D**) The circadian amplitude of locomotor activity rhythms (Qp values). Values are mean ± SEM; *n* = 9 for control, *n* = 9 for *Vip-CK1δ⁻/⁻* mice. ns, not significant. The underlying data can be found in S1 Data. See also S2 Fig.

## PER2::LUC oscillations in SCN slices partially recapitulate lengthened behavioral periods of *CaMKIIα-CK1δ⁻/⁻* and *Avp-CK1δ⁻/⁻* mice

To evaluate the status of cellular circadian clocks in the SCN, we next performed real-time bioluminescent cell imaging of coronal SCN slices prepared from control, *Avp-CK1δ⁻/⁻*, *CaMKIIα-CK1δ⁻/⁻*, and *Vip-CK1δ⁻/⁻* adult mice crossed with a luciferase reporter (*Per2::Luc*) [35] housed in LD. In the previous study, we reported that PER2::LUC oscillations in the SCN slices of *Avp-CK1δ⁻/⁻* mice did not demonstrate coherent circadian rhythm with a lengthened period, contrary to their locomotor activity rhythm [23].

Here, we monitored and compared the day-by-day change of individual pixels' periods of PER2::LUC oscillations between shell and core, as well as among genotypes. To do so, we calculated the daily peak phase and period (the interval of 2 adjacent peaks) for PER2::LUC oscillations in individual pixels that covered the SCN (Figs 3 and S3). In *Avp-CK1δ⁻/⁻; Per2::Luc*

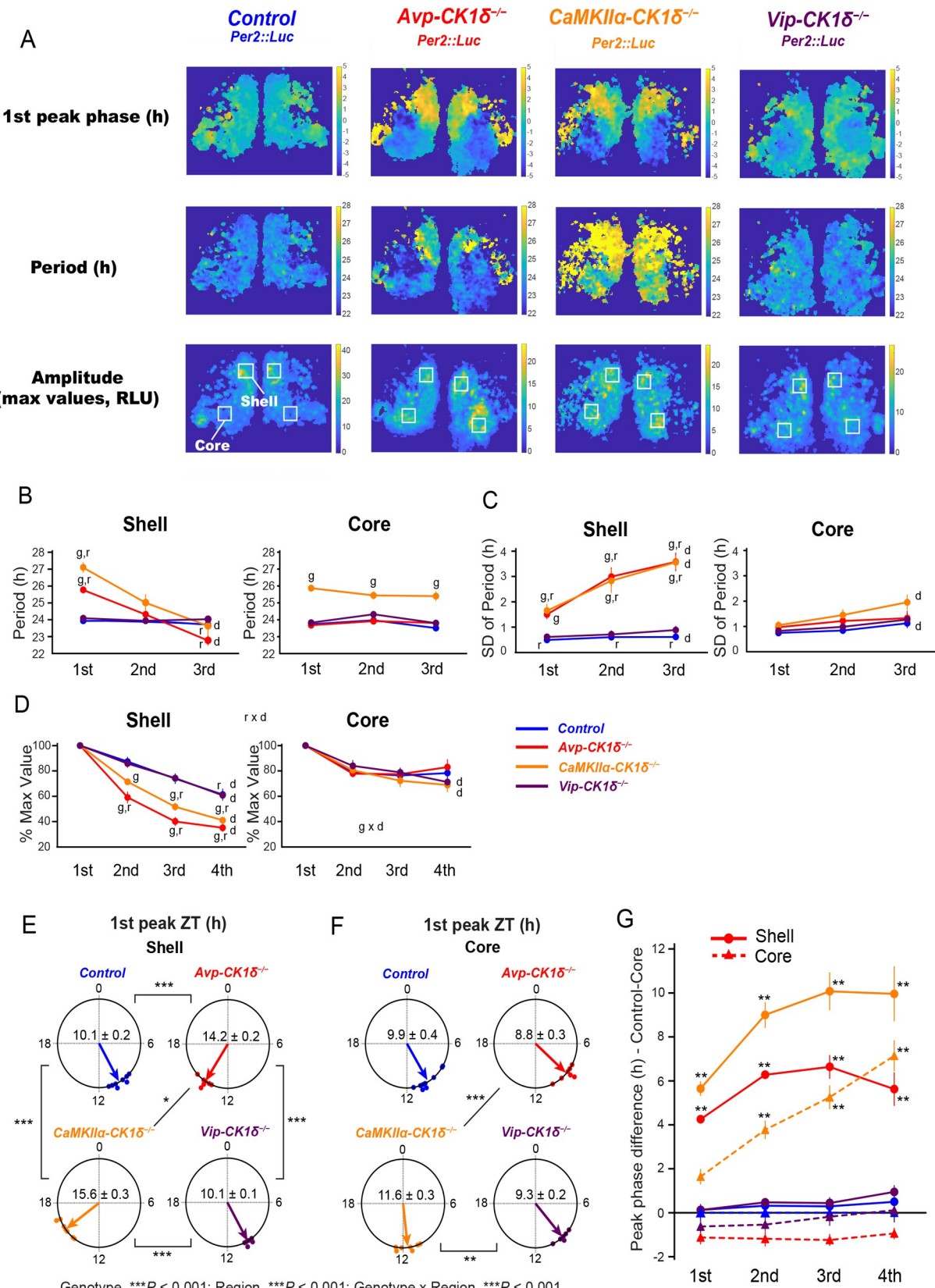

Genotype, ***P < 0.001; Region, ***P < 0.001; Genotype x Region, ***P < 0.001

**Fig 3. Periods of PER2::LUC oscillations in the SCN shell rapidly changes in the SCN slices of *Avp-CK1δ*$^{-/-}$ and *CaMKIIα-CK1δ*$^{-/-}$ mice.** (**A**) Representative first peak phase (relative phase to the slice mean), period (the interval between the first and second peaks), and amplitude maps of PER2::LUC oscillation at the pixel level in coronal SCN slices prepared from control, *Avp-CK1δ*$^{-/-}$, *CaMKIIα-CK1δ*$^{-/-}$, and *Vip-CK1δ*$^{-/-}$ mice. Peak phases, periods, and amplitudes of PER2::LUC oscillations in the individual pixels covering the SCN were calculated for every cycle. White squares on the amplitude maps indicate regions (15 × 15 pixels) considered as the shell or core for further analyses. (**B**) The day-by-day change in the mean period of individual pixels' PER2::LUC oscillations in the shell (left) and core (right) regions. The analyzed regions' examples are shown in Fig 3A. Definitions of peaks, periods, and cycles are indicated in S3 Fig. The averaged values of 2 regions in the left and right SCN of individual slices were considered the representatives of individual mice. Blue, Control; Red, *Avp-CK1δ*$^{-/-}$; Orange, *CaMKIIα-CK1δ*$^{-/-}$; Purple, *Vip-CK1δ*$^{-/-}$. (**C**) The day-by-day change in the standard deviation (SD) of individual pixels' periods in the shell (left) and core (right) regions. (**D**) The day-by-day change in the mean peak amplitude (relative to the first peak amplitude) of individual pixels' PER2::LUC oscillations in the shell (left) and core (right) regions. (**E, F**) The mean first peak phase (ZT) of the shell (**E**) and core (**F**) regions in different mouse lines were shown as Rayleigh plots. Individual dots indicate the mean peak phases of each mouse. (**G**) Peak phase differences between the SCN core of control mice and the shell or core of other mouse lines. Circle, shell; triangles, core. Values are mean ± SEM; n = 9 for Control, n = 7 for *Avp-CK1δ*$^{-/-}$, n = 7 for *CaMKIIα-CK1δ*$^{-/-}$, n = 6 for *Vip-CK1δ*$^{-/-}$. The underlying data can be found in S1 Data. Letters indicate significant differences in the factor of genotype (g: compared with Control), day (d), or region (r). g: effect of genotype, $P < 0.05$ by Kruskal–Wallis rank sum test followed by Mann–Whitney *U* test with Bonferroni correction (**B, C**) or three-way repeated measures ANOVA with post hoc Ryan test (**D**); d: effect of day, $P < 0.05$ by Friedman rank sum test followed by Wilcoxon signed rank test with Bonferroni correction (**B, C**) or three-way repeated measures ANOVA with post hoc Ryan test (**D**); r: effect of region, $P < 0.05$ by Wilcoxon signed rank test (**B, C**) or three-way repeated measures ANOVA with post hoc Ryan test (**D**); g × d, interaction between genotype and day (**D, Core**); r × d, interaction between region and day (**D, in all genotypes**). *$P < 0.05$; **$P < 0.01$; ***$P < 0.001$ by Harrison–Kanji test followed by Watson–Williams test with Bonferroni correction (**E, F**) or Kruskal–Wallis rank sum test followed by Mann–Whitney *U* test with Bonferroni correction (**G**, effect of Genotype-Region compared with Control-Core). *P* values of Rayleigh test were < 0.01 for all circular data. See also S3 Fig.

mice, the peak phases in the shell pixels were later than those in core pixels by approximately 5 h at the first peak (Fig 3A and 3E–3G). Also, the periods were longer in the shell (25.77 ± 0.21 h) than in the core (23.93 ± 0.13 h) by approximately 2 h for the first cycle (Fig 3A and 3B). While core pixels oscillated with relatively stable periods, those of shell pixels shortened rapidly in the subsequent cycles, as if core cells influenced the periods of shell cells (Figs 3B and S3). The results of *Avp-CK1δ*$^{-/-}$; *Per2*::*Luc* mice were consistent with our previous report [23].

In *CaMKIIα-CK1δ*$^{-/-}$; *Per2*::*Luc* mice, all SCN neurons appear to have similarly lengthened periods of PER2::LUC rhythms. Therefore, we expected that SCN explants of those mice would show coherent PER2::LUC oscillations similar to those of control mice, except for the lengthening of periods. Indeed, core pixels oscillated stably with lengthened periods (approximately 25.5 h) for 3 cycles (Fig 3B, right). However, we observed a rapid temporal change of periods in the shell pixels similar to that in *Avp-CK1δ*$^{-/-}$; *Per2*::*Luc* mice, except for approximately 1-h lengthening (Fig 3B, left). Namely, the peak phases in the shell pixels were later than those in core pixels by approximately 4 h at the first peak (Fig 3A and 3E–3G). In addition, the periods were longer in the shell (27.09 ± 0.32 h) than in the core (25.87 ± 0.11 h) by approximately 1 h for the first cycle. However, periods of shell pixels shortened rapidly in the subsequent cycles (Fig 3B). Thus, the coherent circadian rhythm was not maintained in PER2::LUC expression of the SCN slice cultures in *CaMKIIα-CK1δ*$^{-/-}$ mice, which is a common phenomenon with *Avp-CK1δ*$^{-/-}$ mice. Concordantly, the variability of the individual pixels' periods increased in the SCN shell of *Avp-CK1δ*$^{-/-}$ and *CaMKIIα-CK1δ*$^{-/-}$ mice (Fig 3C). Furthermore, the peak amplitude of individual pixels' PER2::LUC oscillations decayed more rapidly in the shell of these 2 strains of mice (Fig 3D). These observations suggested the attenuated synchronization of cellular PER2::LUC oscillations in the SCN shell of *Avp-CK1δ*$^{-/-}$ and *CaMKIIα-CK1δ*$^{-/-}$ mice ex vivo. In contrast, PER2::LUC oscillations in the SCN slices of *Vip-CK1δ*$^{-/-}$; *Per2*::*Luc* mice demonstrated no significant difference compared to control mice. This observation was consistent with their normal free-running period of circadian locomotor activity rhythm.

## The circadian period of calcium rhythm in SCN AVP neurons of *Avp-CK1δ*$^{-/-}$ mice is stably lengthened in vivo

Lengthening the cellular circadian periods of AVP neurons could stably lengthen the free-running period of the behavioral rhythm. However, it could not lengthen the cellular PER2::LUC

rhythms in the prolonged SCN cultures. Therefore, we postulated that the action of AVP neurons might be susceptible to slicing. To test this possibility, we next measured the cellular period of SCN AVP neurons in vivo by recording the intracellular $Ca^{2+}$ ($[Ca^{2+}]_i$) rhythm using fiber photometry [36] in *Avp-CK1δ*$^{-/-}$ and control (*Avp-Cre; CK1δ*$^{wt/flox}$) mice (Fig 4). To do so, we expressed a fluorescent $Ca^{2+}$ indicator jGCaMP7s [37] specifically in SCN AVP neurons by focally injecting a Cre-dependent AAV vector (AAV-*CAG-DIO-jGCaMP7s*) and then implanted an optical fiber just above the SCN (Fig 4A and 4B). The detected jGCaMP7s signal was averaged, detrended, and smoothened to extract daily $[Ca^{2+}]_i$ rhythms (see Materials and methods). We defined the time points crossing zero values as the onset and offset of the GCaMP signal and their midpoint as the peak phase. In both control and *Avp-CK1δ*$^{-/-}$ mice, daily $[Ca^{2+}]_i$ rhythms were observed in SCN AVP neurons in both LD and DD. Their peaks were around the offset of locomotor activity (home-cage activity) (Figs 4C, 4D, 5A and S4A), as described previously for control mice [36]. The relationship between the GCaMP peak and locomotor offset remained unchanged in *Avp-CK1δ*$^{-/-}$ mice. Thus, not only the period of locomotor activity rhythm but also the period of $[Ca^{2+}]_i$ rhythm in AVP neurons was lengthened similarly in DD (GCaMP, control 23.77 ± 0.18 versus *Avp-CK1δ*$^{-/-}$ 24.42 ± 0.13, $P < 0.05$, Fig 5D; behavior, control 23.77 ± 0.10 versus *Avp-CK1δ*$^{-/-}$ 24.43 ± 0.09, $P < 0.001$, Fig 5F, black). Such a lengthening of $[Ca^{2+}]_i$ rhythm period was consistently observed from the onset of DD (Fig 5H). This result indicated that the *CK1δ* deletion in AVP neurons caused a stable lengthening of their cellular circadian period in vivo.

## AVP neurons control the calcium rhythm of VIP neurons in vivo

We next asked whether the SCN shell and core circadian oscillations in vivo in *Avp-CK1δ*$^{-/-}$ mice were dissociated as in the slices or maintained synchrony. Because of the importance of VIP neurons in the SCN core for maintaining the coherence of circadian oscillators in the SCN [14,15,18,38,39], we examined whether the $[Ca^{2+}]_i$ rhythm of SCN VIP neurons was altered in *Avp-CK1δ*$^{-/-}$ mice in vivo. To specifically target the jGCaMP7s expression in VIP neurons of *Avp-CK1δ*$^{-/-}$ mice, we first crossed *Avp-CK1δ*$^{-/-}$ (or control) with *Vip-tTA* knock-in mice [40]. In *Vip-tTA* mice, VIP neurons specifically express tetracycline transactivator (tTA). We then injected a tTA-dependent AAV vector (AAV-*TRE-jGCaMP7s*) into the SCN (Fig 6A and 6B). The $[Ca^{2+}]_i$ rhythms of VIP neurons were synchronized antiphasically with locomotor activity rhythm in both control (*Avp-Cre;CK1δ*$^{wt/flox}$; *Vip-tTA*) and *Avp-CK1δ*$^{-/-}$; *Vip-tTA* mice (Figs 6C, 6D and S4B). Furthermore, the timing of locomotor onset and GCaMP offset were almost identical, while the timing of locomotor offset and VIP GCaMP onset correlated in both LD and DD (Figs 6C, 6D and S4B). In *Avp-CK1δ*$^{-/-}$ mice, the period of VIP-neuronal $[Ca^{2+}]_i$ rhythm was lengthened in DD to the same extent as AVP-neuronal $[Ca^{2+}]_i$ and behavior rhythms (GCaMP, control 23.87 ± 0.03 versus *Avp-CK1δ*$^{-/-}$ 24.47 ± 0.05, $P < 0.001$, Fig 5E). Such a lengthening of $[Ca^{2+}]_i$ rhythm period was consistently observed from the onset of DD (Fig 5I). Importantly, as a control, we verified that EGFP-expressing VIP neurons via injecting AAV-*TRE-EGFP* showed very little circadian oscillation of fluorescence (S4C Fig). These results suggested that the cellular circadian rhythms of AVP and VIP neurons maintained synchrony in vivo in *Avp-CK1δ*$^{-/-}$ mice.

How about the phase relationships between AVP-neuronal, VIP-neuronal, and behavioral circadian rhythms? Overall, the peak phases of GCaMP signals in LD were similar between control and *Avp-CK1δ*$^{-/-}$ mice for both AVP and VIP neurons (AVP: control, ZT 2.74 ± 0.57; *Avp-CK1δ*$^{-/-}$, ZT 3.82 ± 0.74, Fig 5A, top; VIP: control, ZT 4.98 ± 0.54; *Avp-CK1δ*$^{-/-}$, ZT 5.99 ± 0.17, Fig 5A, bottom). AVP neurons peaked earlier than VIP neurons (control, $P < 0.05$; *Avp-CK1δ*$^{-/-}$, $P = 0.07$) in LD. These peak phase relationships were essentially

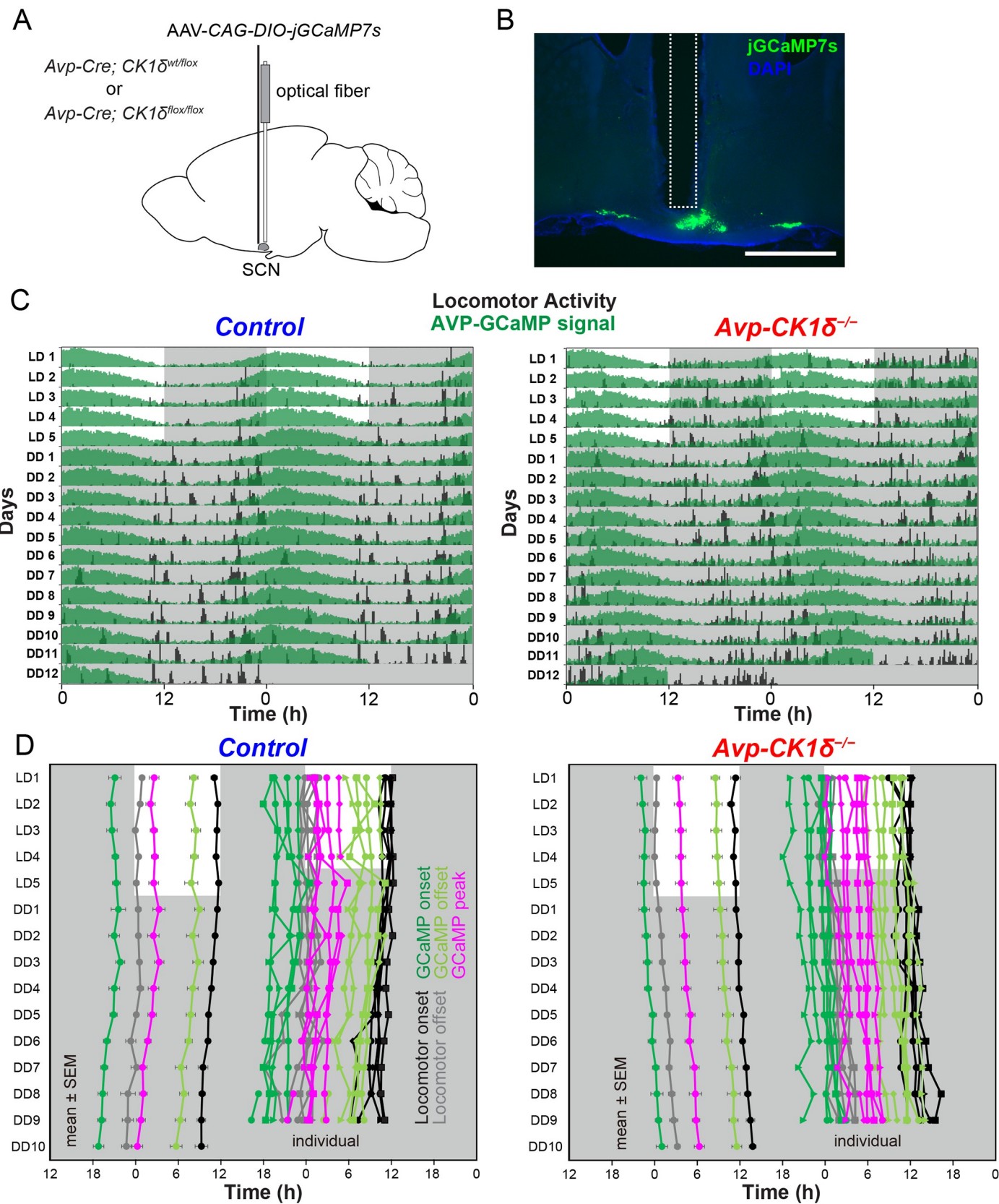

**Fig 4. The in vivo circadian period of AVP-neuronal [Ca$^{2+}$]$_i$ rhythm is lengthened in the SCN of *Avp-CK1δ*$^{-/-}$ mice.** (**A**) Schematic diagram of viral vector (AAV-*CAG-DIO-jGCaMP7s*) injection and optical fiber implantation at SCN in control (*Avp-Cre; CK1δ*$^{wt/flox}$) or *Avp-CK1δ*$^{-/-}$ (*Avp-Cre; CK1δ*$^{flox/flox}$) mice for fiber photometry recording. (**B**) A representative coronal section of mice with jGCaMP7s expression in SCN AVP neurons. A white dotted square shows the estimated position of implanted optical fiber. Green, jGCaMP7s; blue, DAPI. Scale bar, 1 mm. (**C**) Representative plots of the in vivo jGCaMP7s signal of SCN AVP neurons (green) overlaid with locomotor activity (home-cage activity) (black) in actograms. Control (Left) and *Avp-CK1δ*$^{-/-}$ (Right) mice were initially housed in LD (LD1 to LD5) and then in DD (DD1 to DD12). The dark periods are represented as gray shaded areas. (**D**) Plots of locomotor activity onset (black), activity offset (gray), GCaMP onset (green), GCaMP offset (light green), and GCaMP peak (magenta) of mean ± SEM (left column) and individual mice data (right column) in control and *Avp-CK1δ*$^{-/-}$ mice. Identical marker shapes indicate data from the same animal. *n* = 5 for control (*n* = 3 without *Vip-tTA*; *n* = 2 with *Vip-tTA*), *n* = 6 for *Avp-CK1δ*$^{-/-}$ (*n* = 4 without *Vip-tTA*; *n* = 2 with *Vip-tTA*). The underlying data can be found in S1 Data. See also S4 Fig.

maintained even in DD. Due to the lengthening of the free-running period, the peak phase of AVP-neuronal [Ca$^{2+}$]$_i$ in projected ZT was significantly later in *Avp-CK1δ*$^{-/-}$ during days 8 to 10 in DD (control, projected ZT 0.72 ± 0.69; *Avp-CK1δ*$^{-/-}$, projected ZT 6.15 ± 0.65, *P* < 0.001, Fig 5B, top) but not different in CT (circadian time) defined by the onset of locomotor activity as CT12 (control, CT 3.16 ± 0.66; *Avp-CK1δ*$^{-/-}$, CT 4.43 ± 0.76, Fig 5C, top). The situation of VIP-neuronal [Ca$^{2+}$]$_i$ in DD was similar (control, projected ZT 4.48 ± 0.22; *Avp-CK1δ*$^{-/-}$, projected ZT 9.47 ± 0.36, *P* < 0.001, Fig 5B, bottom; control, CT 5.64 ± 0.17; *Avp-CK1δ*$^{-/-}$, CT 6.23 ± 0.44, Fig 5C, bottom). Thus, the phase relationship between [Ca$^{2+}$]$_i$ rhythms of AVP and VIP neurons remained unaltered in *Avp-CK1δ*$^{-/-}$ mice. Note that the TTFL period was artificially lengthened only in AVP but not in VIP neurons. These results suggested that AVP neurons can regulate the period of cellular circadian rhythm of VIP neurons in vivo.

## AVP is not necessary for the transmission of the cellular circadian period of AVP neurons

To test whether the AVP peptide is essential for SCN AVP neurons to convey their cellular period to other SCN neurons, we selectively disrupted the *Avp* gene in these neurons using in vivo genome editing. First, we further crossed *Avp-CK1δ*$^{-/-}$ mice with *Rosa26-LSL-Cas9-2A-EGFP* mice, which express SpCas9 Cre-dependently [41]. We then injected an AAV vector expressing gRNA targeting the *Avp* (AAV-*U6-gAvp-EF1α-DIO-mCherry*) or a control AAV (AAV-*U6-gControl-EF1α-DIO-mCherry*) into the SCN of *Avp-CK1δ*$^{-/-}$; *Rosa26-LSL-Cas9* mice (S5A Fig) [41,42]. Injecting AAV-*U6-gAvp-EF1α-DIO-mCherry* efficiently reduced AVP immunoreactivity in the SCN (S5B–S5F Fig). However, the *Avp* knockdown did not significantly reduce the lengthening of the free-running period of these mice (S5G–S5I Fig). Thus, AVP peptide may be dispensable for transmitting the period length of the cellular clocks of AVP neurons to the entire SCN network.

## VIP neurons respond to the optogenetic activation of AVP neurons in vivo

To regulate the calcium rhythm of VIP neurons, SCN AVP neurons should have some functional connectivity with VIP neurons. Therefore, we tested the projection of AVP neurons to VIP neurons by anterograde tracing. To do so, we injected an AAV vector expressing the Synaptophysin-fused-GFP reporter in a Cre-dependent manner (AAV-*EF1α-DIO-Synaptophysin::GFP*) [43] into the SCN of *Avp-Cre* mice. Synaptophysin::GFP-positive fibers of SCN AVP neurons were scattered throughout the SCN (Fig 7A). We confirmed at least some, but not many, appositions of these fibers onto VIP neurons (Fig 7B). This result was consistent with a previous study reporting that AVP fibers make sparse contacts onto VIP neurons, whereas they make numerous contacts onto GRP and calretinin neurons; the latter two, in turn, make dense contacts onto VIP neurons [44].

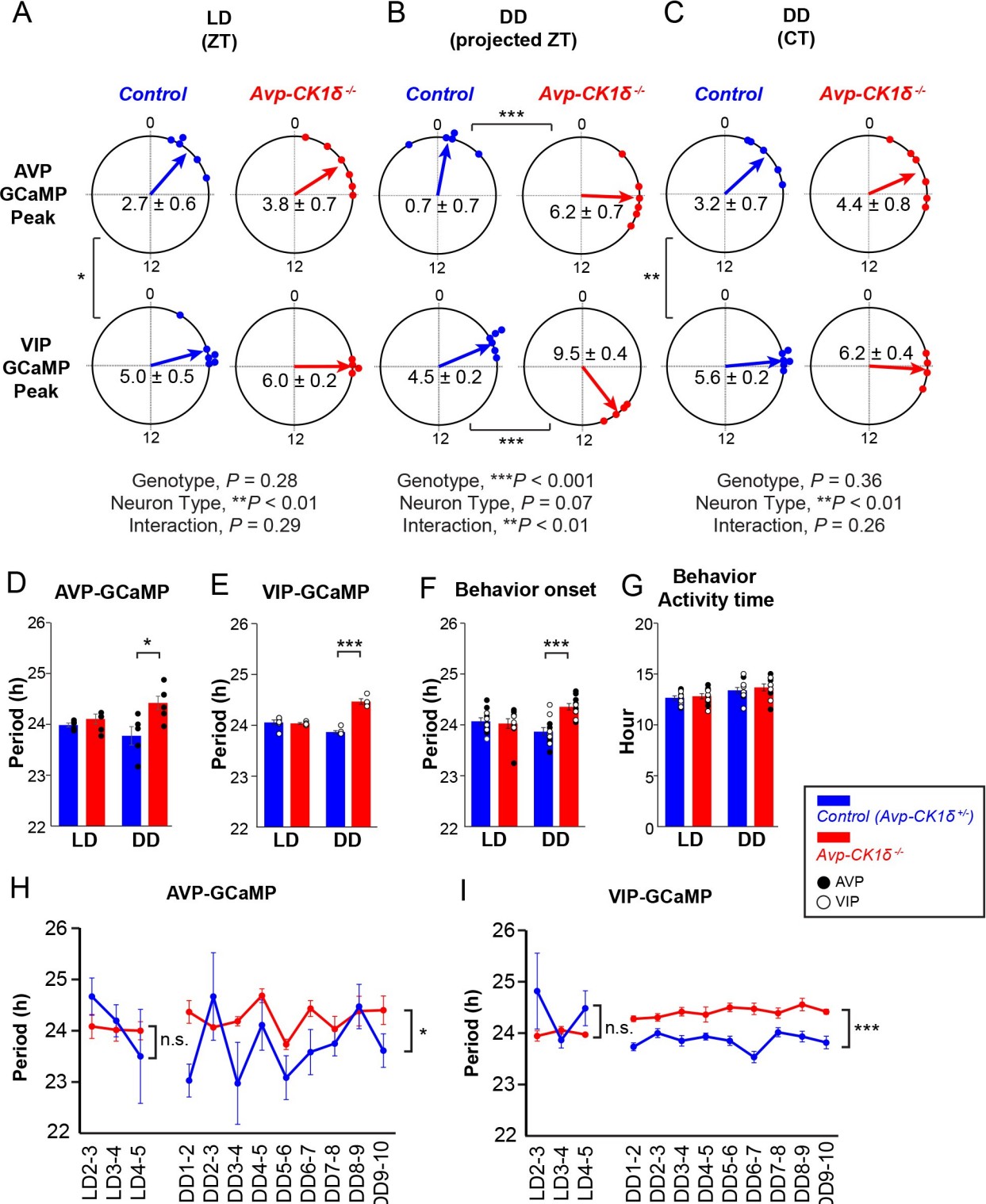

**Fig 5. Periods of AVP-neuronal, VIP-neuronal, and behavior rhythms are similarly lengthened in *Avp-CK1δ⁻/⁻* mice in vivo.** (A-C) Peak phases of AVP and VIP GCaMP fluorescence rhythms in LD (LD1-5, **A**) or DD in projected ZT (DD8-10, **B**) or CT (**C**) were shown as Rayleigh plots. Individual dots indicate the peak phases of each mouse. (**D**) Periods of AVP-neuronal GCaMP fluorescence rhythm in LD (LD1-5, left) or DD (DD8-10, right). (**E**) Periods of VIP-neuronal GCaMP fluorescence rhythm in LD or DD. (**F**) Periods of locomotor activity onset in LD or DD. Black circle, data from AVP-GCaMP experiment; white circle, data from VIP-GCaMP experiment. Data from AVP-GCaMP and VIP-GCaMP experiments were

combined for statistical analysis. (**G**) Activity time of locomotor activity rhythm in LD (LD1-5, left) or in DD (DD8-10, right). Blue, Control (*Avp-CK1δ*$^{+/-}$, i.e., *Avp-Cre; CK1δ*$^{wt/flox}$); red, *Avp-CK1δ*$^{-/-}$. Values are mean ± SEM. *n* = 5 for AVP-GCaMP: Control, *n* = 6 for AVP-GCaMP: *Avp-CK1δ*$^{-/-}$, *n* = 6 for VIP-GCaMP: Control, *n* = 4 for VIP-GCaMP: *Avp-CK1δ*$^{-/-}$. *$P$ < 0.05; **$P$ < 0.01; ***$P$ < 0.001 by Harrison–Kanji test followed by Watson–Williams test (**A** to **C**), or by two-tailed Welch *t* test (**D** to **G**). *P* values of Rayleigh test were < 0.01 for all circular data. (**H**, **I**) Day-by-day changes in the period of AVP- (**H**) or VIP-neuronal (**I**) GCaMP fluorescence rhythms in LD (3 days) and in DD (9 days). Values are mean ± SEM. *n* = 3 for AVP-GCaMP: Control, *n* = 5 for AVP-GCaMP: *Avp-CK1δ*$^{-/-}$, *n* = 5 for VIP-GCaMP: Control, *n* = 3 for VIP-GCaMP: *Avp-CK1δ*$^{-/-}$. *$P$ < 0.05; ***$P$ < 0.001 by two-way repeated measures ANOVA. n.s., not significant. The underlying data can be found in S1 Data.

Lastly, we investigated the functional connectivity between AVP and VIP neurons by recording $[Ca^{2+}]_i$ response of VIP neurons to optogenetic activation of SCN AVP neurons in vivo. To this end, we expressed ChrimsonR-mCherry, a red light-gated cation channel [45,46], and jGCaMP7s in AVP and VIP neurons, respectively, by injecting AAV-*CAG-FLEX-ChrimsonR-mCherry* and AAV-*TRE-jGCaMP7s* into the SCN of *Avp-Cre; Vip-tTA* mice (Fig 7C and 7D). Strikingly, optogenetic activation of SCN AVP neurons around ZT22, when $[Ca^{2+}]_i$ in AVP and VIP neurons is relatively low, acutely increased jGCaMP7s signal in VIP neurons as measured by fiber photometry (Fig 7E and 7F). These data suggest that SCN AVP neurons can activate VIP neurons in vivo.

## Discussion

In the present study, we showed that the period lengthening of the behavior rhythm caused by pan-SCN deletion of *CK1δ* was comparable to that caused by AVP neuron–specific deletion. In contrast, we did not observe any change in the behavioral period in mice with VIP neuron–specific *CK1δ* deficiency. In slices, PER2::LUC rhythms did not fully reproduce the presumed lengthening of the ensemble period of SCN network in *Avp-CK1δ*$^{-/-}$ mice. In contrast, in vivo, $[Ca^{2+}]_i$ rhythms in both SCN AVP and VIP neurons oscillated coherently in *Avp-CK1δ*$^{-/-}$ mice with a lengthened period along with the behavior rhythm. In addition, optogenetic stimulation of AVP neurons acutely increased $[Ca^{2+}]_i$ in VIP neurons. Collectively, these data suggest that AVP neurons play a primary role in the circadian pacesetting in vivo to determine the ensemble period of the SCN network. This observation highlights the importance of in vivo analyses of SCN network dynamics.

### AVP neurons are the principal determinant of the ensemble period of SCN in vivo

Previous studies utilizing neuron type–specific genetic manipulations of cellular periods have suggested the importance of NMS neurons, AVP neurons, Drd1a neurons, and VPAC2 neurons in setting the ensemble period of the SCN [21,23,24,25,47]. In contrast, although VIP is a critical synchronizer for SCN neurons, the TTFL in VIP neurons itself may contribute little to the pacesetting of the entire SCN network.

Recently, Hamnett and colleagues demonstrated that lengthening cellular periods in VPAC2 neurons, which were marked by a specific Cre driver line, also lengthens the free-running period of the circadian wheel-running rhythm [25]. However, PER2::LUC rhythms of SCN slices did not reflect the behavioral period of these mice. Such inconsistency between in vivo and ex vivo periods is similar to our current and previous observations for AVP neuron–specific lengthening of cellular periods. Furthermore, these manipulated VPAC2 neurons were mostly confined in the SCN shell and include approximately 85% of AVP neurons [28]. Therefore, their and our findings implicate that the ability of VPAC2/AVP neurons to set the SCN ensemble period in vivo is lost in slices. Interestingly, they reported that VPAC2 neurons additionally require the contribution of VIP neurons to dictate the SCN slice period [28]. This view also explains that the cellular period lengthening in NMS neurons, including both AVP and VIP neurons, successfully lengthened the periods of both SCN slices and behavior [21]. On the

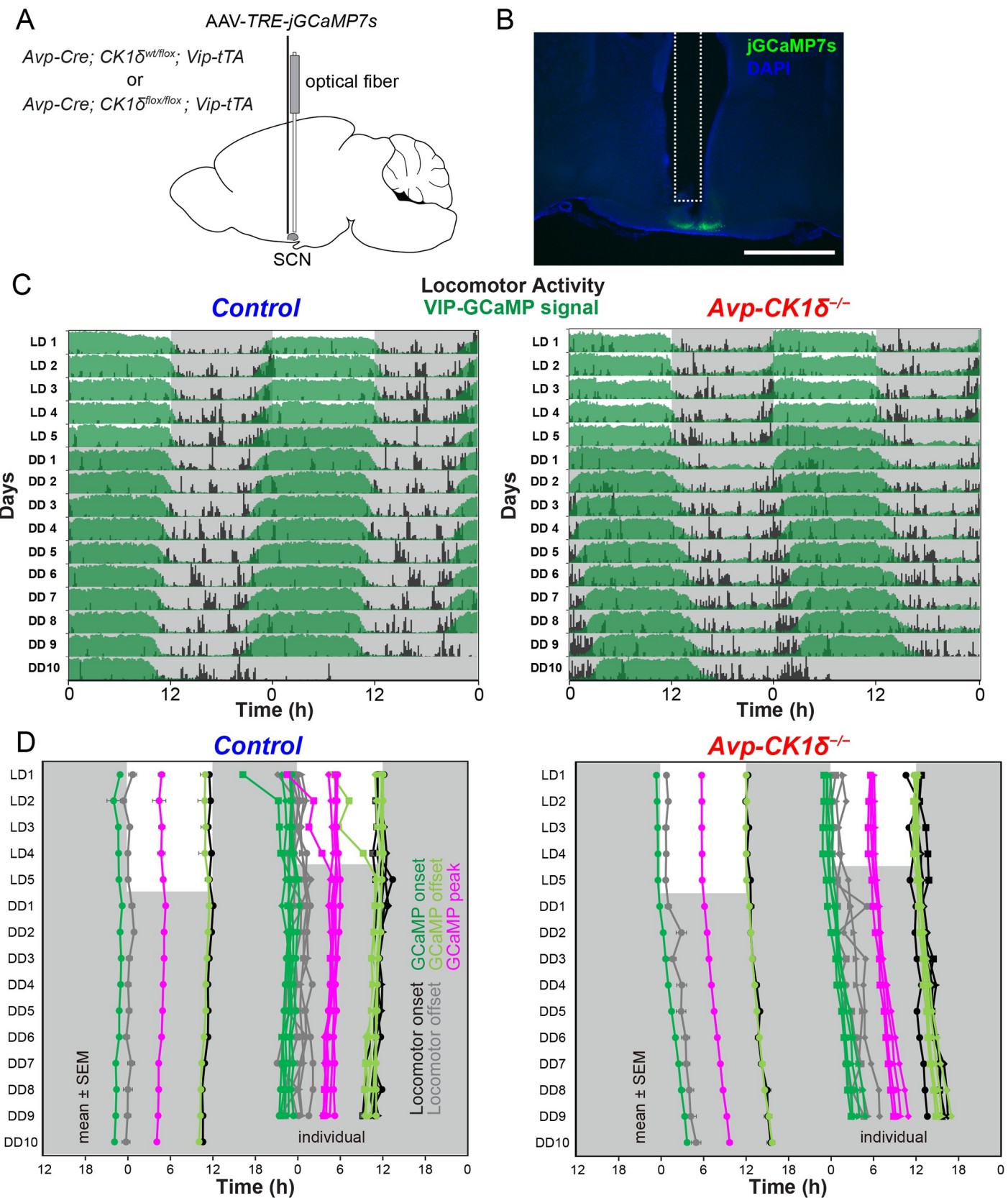

**Fig 6. The in vivo circadian period of VIP-neuronal [Ca$^{2+}$]$_i$ rhythm is lengthened in the SCN of *Avp-CK1δ$^{−/−}$* mice.** (**A**) Schematic diagram of viral vector (AAV-*TRE-jGCaMP7s*) injection and optical fiber implantation at SCN in control (*Avp-Cre; CK1δ$^{wt/flox}$; Vip-tTA*) or *Avp-CK1δ$^{−/−}$* (*Avp-Cre; CK1δ$^{flox/flox}$; Vip-tTA*) mice for fiber photometry recording. (**B**) A representative coronal section of mice with jGCaMP7s expression in SCN VIP neurons. A white dotted square shows the estimated position of implanted optical fiber. Green, jGCaMP7s; blue, DAPI. Scale bar, 1 mm. (**C**) Representative plots of the in vivo jGCaMP7s signal of SCN VIP neurons (green) overlaid with locomotor activity (home-cage activity) (black) in actograms. Control (Left) and *Avp-CK1δ$^{−/−}$* (Right) mice were initially housed in LD (LD1 to LD5) and then in DD (DD1 to DD10). The dark periods are represented as gray shaded areas. (**D**) Plots of locomotor activity onset (black), activity offset (gray), GCaMP onset (green), GCaMP offset (light green), and GCaMP peak (magenta) of mean ± SEM (left column) and individual mice data (right column) in control and *Avp-CK1δ$^{−/−}$* mice. Identical marker shapes indicate data from the same animal. *n* = 6 for control, *n* = 4 for *Avp-CK1δ$^{−/−}$*. The underlying data can be found in S1 Data. See also S4 Fig.

other hand, an essential significance of our current study is the demonstration that in vivo AVP neurons alone can control the circadian rhythms of other SCN neurons, such as VIP neurons, and determine the SCN ensemble period to control behavior rhythm.

Indeed, it was recently reported that the optogenetic stimulation of SCN AVP neurons advances the circadian behavior rhythm [48]. In this study, we also demonstrated that the optogenetic stimulation of these neurons immediately increased VIP neuronal [Ca$^{2+}$]$_i$ in vivo. However, whether this effect is mediated by a direct or indirect connection of AVP neurons to VIP neurons remains elusive. An SCN connectome analysis reported only sparse projections of AVP neurons directly onto VIP neurons [44]. Instead, most GRP and calretinin neurons receive AVP fibers and send projections to VIP neurons in the SCN. Taken together, AVP neurons may functionally connect to VIP neurons indirectly via GRP or calretinin neurons.

## The stability of the SCN circadian period differs between ex vivo and in vivo

So, what made the difference in periods between in vivo and ex vivo? AVP neurons may require direct or indirect connections with extra-SCN regions, lost in slices, to exert influence on the entire SCN. Another possibility, which seems more appealing to us, is that slicing damaged the usual connections and humoral environment inside the SCN so that AVP neurons are unable to transmit period information to other SCN neurons. It is also formally possible that the difference is due to more general factors, such as PER2::LUC rhythm resetting by culture and lower survival of *CK1δ*-deficient cells in slices. Another technical limitation of this study may be the methodological difference in how the period was calculated between ex vivo (PER2::LUC) and in vivo ([Ca$^{2+}$]$_i$) measurements. Although Ca$^{2+}$ imaging has been successfully performed in neonatal organotypic SCN cultures [49,50], we are unaware of any Ca$^{2+}$ imaging studies in adult SCN cultures. Similarly, techniques to measure the PER2::LUC rhythm of VIP neurons in *Avp-CK1δ$^{−/−}$* mice in vivo are currently unavailable. Nevertheless, [Ca$^{2+}$]$_i$ and PER2::LUC are likely to oscillate with the same period but different phases most of the time [49,50].

We previously showed that GABA from AVP neurons is unnecessary for transmitting their cellular period to the behavioral period [36]. Here, we show that AVP is also dispensable for the circadian pacesetting in *Avp-CK1δ$^{−/−}$* mice. This finding is consistent with previous reports that genetic and pharmacological blockade of AVP signaling may attenuate interneuronal communication in the SCN but causes little change in the period of behavior or SCN slice PER2::LUC rhythms [51–55], with the exception of one study reporting that high doses of V1a or V1b receptor antagonists lengthened the PER2::LUC periods in SCN slices [56]. However, it is still possible that AVP and GABA act as redundant transmitters to mediate the AVP cellular period. Alternatively, other peptides expressed in SCN AVP neurons, such as NMS, Cholecystokinin, and Prokineticin 2 [21,57–59], may be responsible for the ensemble period-setting. In general, peptides are often engaged in volume transmission rather than synaptic transmission [60]. Therefore, the peptidergic transmission by AVP neurons might be more diffusible and attenuated in slice cultures. Also, because AVP neurons are distributed around the core, the ratio of AVP neurons to core cells was smaller in coronal sections containing both shell

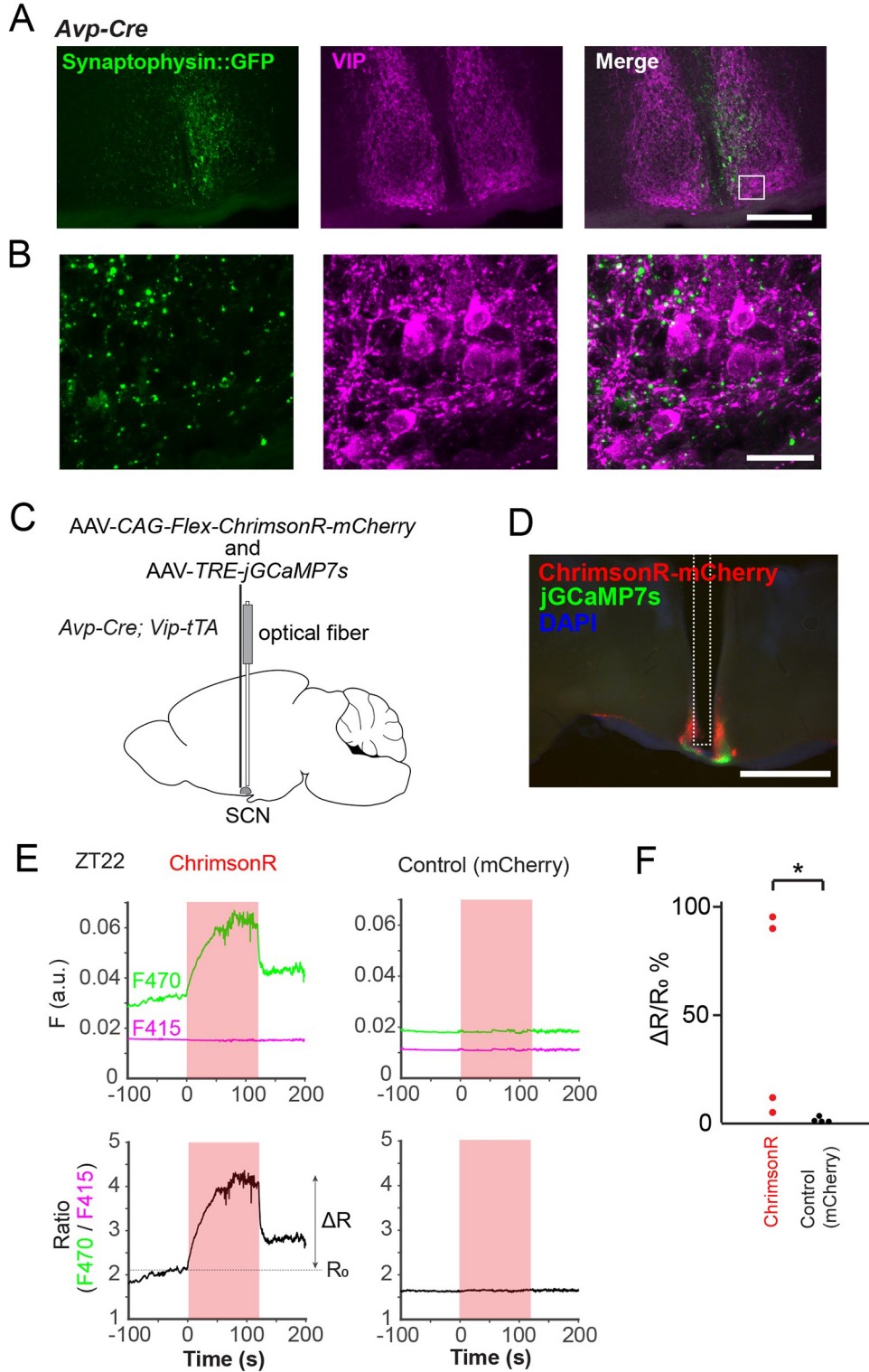

**Fig 7. AVP neurons functionally connect to VIP neurons in the SCN.** (**A**, **B**) The fusion protein Synaptophysin::GFP is expressed in the axon terminals of AVP neurons by injecting AAV-*EF1α-DIO-synaptophysin*::*GFP* into the SCN of an *Avp-Cre* mouse. From the left, a representative coronal section of the SCN showing the native Synaptophysin::GFP fluorescence (green), VIP immunostaining (magenta), or the merged signals (white). The white rectangle in (**A**) indicates the position of the magnified images of (**B**). Some Synaptophysin::GFP-labeled AVP neuron axon terminals

are present around VIP neurons in the SCN. Scale bars, 200 (**A**) or 20 μm (**B**). (**C**) Schematic diagram of viral vector (AAV-*CAG-Flex-ChrimsonR-mCherry* and AAV-*TRE-jGCaMP7s*) injection and optical fiber implantation at the SCN in *Avp-Cre; Vip-tTA* mice for fiber photometry recording of SCN VIP-neuronal $[Ca^{2+}]_i$ and optogenetic stimulation of SCN AVP neurons. (**D**) A representative coronal section of mice with ChrimsonR-mCherry expression in SCN AVP neurons and jGCaMP7s expression in SCN VIP neurons. A white dotted square indicates the estimated position of the implanted optical fiber. Red, mCheery; green, jGCaMP7s; blue, DAPI. Scale bar, 1 mm. (**E**) Top: Representative traces of the jGCaMP7s signal of SCN VIP neurons upon optogenetic stimulation of AVP neurons at ZT22 in vivo (left, ChrimsonR; right, mCherry Control). Green traces indicate the fluorescence (**F**) value at the 470-nm light excitation (F470), $Ca^{2+}$-dependent signal. Magenta traces indicate the fluorescence value at the 415-nm light excitation (F415), $Ca^{2+}$-independent control signal. Red shading indicates the timing of optical stimulation (635 nm, 50 ms pulse, 5 Hz, 120 s). Bottom: Ratio (R) calculated by F470/F415 from the upper traces. The baseline ratio ($R_0$) is the mean R-value of the prestimulation period (−30 s–0 s). ΔR is the difference between the mean R-value of the late phase during the stimulation period (90 s–120 s) and $R_0$. a.u., arbitrary unit. Optogenetic stimulation of SCN AVP neurons increases VIP-neuronal $[Ca^{2+}]_i$ in freely moving mice. (**F**) Comparison of the $\Delta R/R_0$% during the late stimulation period. $n = 4$. *$P < 0.05$ by Mann–Whitney *U* test.

and core than in vivo. Such a condition may be another reason why the cellular period of AVP neurons was not reflected throughout the SCN in slices.

In this context, it was surprising that PER2::LUC rhythms were dissociated between shell and core in the SCN slices of *CaMKIIα-CK1δ$^{-/-}$* mice. Because, in contrast to *Avp-CK1δ$^{-/-}$* mice, all SCN neurons were supposed to lack *CK1δ*. A simple explanation for this observation is that *CK1δ* contributes to the cellular period-setting differently between shell and core cells. Its deficiency might lengthen the cellular period more in the shell than in the core, resulting in chimerism of the cellular periods within the SCN, as in *Avp-CK1δ$^{-/-}$* mice. It is technically challenging to precisely measure the intrinsic cellular period of each type of *CK1δ$^{-/-}$* SCN neurons (e.g., AVP neurons and VIP neurons) under conditions in which extracellular inputs are completely excluded [10,11,61]. The initial periods of PER2::LUC rhythms in the culture might reflect but were not the same as the intrinsic cellular periods. This is because there should be some intercellular communications that could affect the cellular periods.

The lengthening of the behavioral periods by approximately 0.8 to 0.9 h in *Avp-CK1δ$^{-/-}$* and *CaMKIIα-CK1δ$^{-/-}$* mice seemed to be less than the approximately 1.5- to 2-h lengthening observed in *CK1δ$^{-/-}$* peripheral cells, such as primary mouse embryonic fibroblasts (MEFs) and liver explants [8]. Nevertheless, such an apparent reduction of period lengthening in behavior rhythm may be attributed to the nature of in vivo SCN network. Indeed, the circadian period alteration caused by a series of CRY1 mutations was much smaller in behavior rhythms than in cellular oscillations of MEFs, converging close to 24 h in behavior [62].

In any case, this study demonstrated the importance of measuring the dynamics of the SCN network in vivo. Although the slice preparations are beneficial for examining the spatiotemporal organization of cellular clocks in the SCN, there is a large gap between the SCN slice and behavioral rhythms. Indeed, inconsistencies between the behavioral free-running period and the period of slice *Per1-Luc*, *Per1-Gfp*, or PER2::LUC oscillations have also been reported [22,63–67]. In vivo recording of the SCN dynamics will play a pivotal role in filling this gap. By combining genetically modified mice and AAV vectors utilizing Cre/loxP and Tet systems [40], we successfully measured the in vivo cellular oscillations of AVP and VIP neurons separately in mice with AVP neuron–specific gene knockout. Such a dual-targeting strategy would be potent for studying the interaction between 2 different types of cells within the SCN, a small but complicated network consisting of multiple types of neurons and glial cells.

## Shapes of calcium rhythm differ between AVP and VIP neurons in the SCN

In vivo calcium activity of AVP neurons was close to the sine curve (Fig 4C). In contrast, the calcium signal of VIP neurons was close to on and off square pulses (Fig 6C), which nearly

delineated the (subjective) day. Such daily patterns of $[Ca^{2+}]_i$ were consistent with the previous reports [18,36,68]. In single SCN neurons in culture, clock gene expression rhythms are quasi-sinusoidal, whereas the neuronal activity rhythms are often quasi-rectangular in shape. $[Ca^{2+}]_i$ rhythms may be the intermediate and variable, which seems to be regulated by both the inter-cellular neuronal network and the intracellular modulators linking the TTFL to $[Ca^{2+}]_i$ [11,12,61,69–71].

By analogy, AVP-neuronal $[Ca^{2+}]_i$ rhythm may be regulated primarily by the intracellular TTFL mechanism, while neuronal firing may contribute more significantly to VIP-neuronal $[Ca^{2+}]_i$. As discussed previously [28], some mechanisms in addition to the neuronal activity and $Ca^{2+}$ influx, such as cAMP responsive element (CRE)-dependent transcription [72], may be essential for modulating TTFL in VPAC2/AVP neurons. The more rectangular $[Ca^{2+}]_i$ rhythm in VIP neurons is like an on-and-off switch. This shape may be suitable for setting time frames according to the LD cycle for locomotor activity, sleep-wakefulness, and other bodily functions.

### AVP neurons as the primary oscillating part of the SCN network

As discussed above, AVP neurons are likely the principal pacesetter of the SCN network clock. In addition, reassessment of the behavioral rhythm in mice with AVP neuron–specific *Bmal1* deletion (*Avp-Bmal1$^{-/-}$* mice) revealed that their wheel-running rhythm was disrupted similarly to mice with NMS or VPAC2 neuron–specific *Bmal1* deletion [21,25].

Altogether, we propose the following model to integrate findings in the previous reports and this study. AVP neurons function as the primary oscillatory part of the SCN network, determining the period and taking charge of the oscillation itself. However, the cellular clocks of AVP neurons may be sloppy and not necessarily self-sustaining. Therefore, the sustained oscillation of the cellular clocks of AVP neurons may need to be driven by other SCN neurons. VIP neurons likely play this role, couple AVP neuronal population, and transmit information about external light to regulate phases of AVP-cellular rhythms. In this way, VIP is critical for circadian pacemaking. On the other hand, the TTFL of AVP neurons regulates the period of rhythmic VIP release. Thus, the functional differentiation and reciprocal interaction between cell types in the SCN network exert the central clock function.

## Materials and methods

### Ethics statements

All experimental procedures were approved by the Kanazawa University Animal Experiment Committee (approval numbers: AP-173836, AP-224337), the Kanazawa University Safety Committee for genetic recombinant experiments (approval numbers: Kindai 6–2124, 6–2598), the Institutional Animal Care and Use Committee at the Meiji University (approval number: IACUC20-126), and Meiji University Recombinant DNA Experiment Safety Committee (approval number: D-19-4). The study was carried out in compliance with Fundamental Guidelines for Proper Conduct of Animal Experiment and Related Activities in Academic Research Institutions under the jurisdiction of the Ministry of Education, Culture, Sports, Science and Technology (Notice No. 71 of 2006), and Standards relating to the Care and Keeping and Reducing Pain of Laboratory Animals (Notice of the Ministry of the Environment No. 88 of 2006).

### Animals

*Avp-Cre* and *Vip-tTA* mice were reported previously [22,40]. This *Avp-Cre* line is a transgenic mouse harboring a modified BAC transgene, which has an insertion of codon-improved Cre

recombinase gene immediately 5′ to the translation initiation codon of exogenous *Avp* gene in the BAC but without manipulation of the endogenous *Avp* loci in the mouse. Thus, the strain is different from the widely used *Avp-ires2-Cre* (JAX #023530) that shows hypomorphic expression of AVP [73], as well as from another *Avp-ires-Cre* line [74]. *CK1δ^{flox}* (JAX #010487) [8], *Bmal1^{flox}* mice (JAX #007668) [75], *Vip-ires-Cre* (*Vip^{tm1(cre)Zjh}*/J, JAX #010908) [33], and *Rosa26-LSL-Cas9-2A-EGFP* mice (JAX #026175) [41] were obtained from Jackson Laboratory. *CaMKIIα-Cre* (*Camk2a::iCreBAC*) [30] was obtained from the European Mouse Mutant Archive (EMMA #01153). The *Per2::Luc* reporter mice were provided by Dr. Joseph Takahashi [35]. All lines were congenic on C57BL/6: *Avp-Cre* and *Vip-tTA* mice were generated on C57BL/6J; *CK1δ^{flox}* (N>12), *Bmal1^{flox}* (N>9), *Vip-ires-Cre* (N>7), *Per2::Luc* (N>10), and *Rosa26-LSL-Cas9-2A-EGFP* (N>7) were backcrossed to C57BL/6J before use; *CaMKIIα-Cre* was backcrossed to C57BL/6N (N>10), then crossed to *CK1δ^{flox}* and *Per2::Luc*. We compared the conditional knockouts with controls whose genetic backgrounds were comparable. *Avp-Cre*, *CaMKIIα-Cre*, *Vip-ires-Cre*, *Per2::Luc*, *Vip-tTA*, and *Rosa26-LSL-Cas9-2A-EGFP* mice were used in hemizygous or heterozygous condition. We used both male and female mice in our experiments (S1 Data). Whether we pooled data from both sexes or analyzed them separately, the conclusions we reached remained the same. Mice were maintained under a strict 12-h light/12-h dark cycle in a temperature- and humidity-controlled room and fed ad libitum.

## Behavioral analyses

Male and female *CaMKIIα-CK1δ^{−/−}* (*CaMKIIα-Cre; CK1δ^{flox/flox}*) (Fig 1), *Vip-CK1δ^{−/−}* (*Vip-ires-Cre; CK1δ^{flox/flox}*) (Fig 2), and *Avp-CK1δ^{−/−}; Rosa26-LSL-Cas9-2A-EGFP* (*Avp-Cre; CK1δ^{flox/flox}; Rosa26-LSL-Cas9-2A-EGFP*) (S5 Fig) mice, aged 8 to 20 weeks, were housed individually in a cage placed in a light-tight chamber (light intensity was approximately 100 lux). Spontaneous locomotor activity (home-cage activity) was monitored by infrared motion sensors (O'Hara) in 1-min bins as described previously [22]. *CK1δ^{flox/flox}* and *CaMKIIα-Cre; CK1δ^{wt/flox}* littermates were used together as the control for *CaMKIIα-CK1δ^{−/−}* mice. *CK1δ^{flox/flox}* and *Vip-ires-Cre; CK1δ^{wt/flox}* littermates were used as the control for *Vip-CK1δ^{−/−}* mice. Because 2 control lines (*CK1δ^{flox/flox}* and *Cre; CK1δ^{wt/flox}*) for each conditional knockout behaved similarly, we regarded them together as control. Actogram, activity profile, and χ² periodogram analyses were performed via ClockLab (Actimetrics). The free-running period was measured by periodogram for days 10 to 24 (Figs 1 and 2) or the last 10 days in DD (S5 Fig). The activity time was calculated from the daily activity profile (average pattern of activity) of the same 15 days using the mean activity level as a threshold for detecting the onset and the offset of activity time [22]. Wheel-running activity was measured for *Avp-CK1δ^{−/−}* (*Avp-Cre; CK1δ^{flox/flox}*), *Avp-Bmal1^{−/−}* (*Avp-Cre; Bmal1^{flox/flox}*), and control mice as described previously [76]. Briefly, each mouse was housed in a separate cage equipped with a running wheel (12-cm diameter; SANKO, Osaka, Japan). The cages were placed in ventilated boxes; the light intensity at the bottom of the cage was 200 to 300 lux. The number of wheel revolutions was counted in 1-min bins. A chronobiology kit (Stanford Software Systems) and ClockLab software (Actimetrics) were used for data collection and analyses. The free-running period was measured by periodogram for the last 7 days in DD (S2 Fig).

## Bioluminescence imaging

The *Avp-CK1δ^{−/−}*, *CaMKIIα-CK1δ^{−/−}*, and *Vip-CK1δ^{−/−}* mice were further mated with *Per2::Luc* reporter mice [35] (*Avp-Cre; CK1δ^{flox/flox}; Per2::Luc, CaMKIIα-Cre; CK1δ^{flox/flox}; Per2::Luc, Vip-Cre; CK1δ^{flox/flox}; Per2::Luc*) and compared to control mice (*CK1δ^{flox/flox}; Per2::Luc*). Male

and female mice aged 8 to 17 weeks were housed in LD before sampling. Coronal SCN slices of 150 μm were made at ZT8–9 with a linear slicer (NLS-MT; Dosaka-EM). The SCN tissue at the mid-rostrocaudal region was cultured, and its PER2::LUC bioluminescence was imaged every 30 min with an exposure time of 25 min with an EMCCD camera (Andor, iXon3), as described previously [23].

Images were analyzed using ImageJ and MATLAB (MathWorks). The resolution of images was adjusted to 4.6 μm/pixel for further analyses. Bioluminescence values in individual pixels were detrended by subtracting 24-h moving average values and then were smoothened with a 15-point moving average method. The middle of the time points crossing the value 0 upward and downward was defined as peak phases, whereas the maximum value between these 2 time points was defined as amplitude. The intervals between 2 adjacent peak phases were calculated as the periods [22]. The values of the regions lateral to the SCN were regarded as background. Therefore, pixels with values lower than the background were eliminated for further analyses. Periods of individual pixels' oscillation were calculated for every cycle. Then, square regions of interest (ROIs, 15 × 15 pixels) were defined in the shell and core of the left and right SCN. Pixels with a period shorter than 4 h or longer than 40 h were eliminated to calculate the mean period, amplitude, and peak phase. Pixels in the left and right ROIs were first processed separately. Then, the averaged values of these two were considered representatives of individual mice.

## Viral vector and surgery

The AAV-2 ITR containing plasmids *pGP-AAV-CAG-FLEX-jGCaMP7s-WPRE* (Addgene plasmid #104495, a gift from Dr. Douglas Kim and GENIE Project) [37] and *pAAV-TRE-EGFP* (Addgene plasmid #89875, a gift from Dr. Hyungbae Kwon) [77] were obtained from Addgene. *pAAV-TRE-jGCaMP7s* and *pAAV-U6-gAvp-EF1α-DIO-mCherry* were described previously [40,42]. *pAAV-U6-gControl-EF1α-DIO-mCherry* contains spacer sequences from *pX333* (Addgene plasmid #64073, a gift from Dr. Andrea Ventura) instead of gRNA sequences for *Avp*. *pAAV-EF1α-DIO-synaptophysin::GFP* [43] was kindly provided by Dr. Richard D. Palmiter, University of Washington. *pAAV-CAG-FLEX-ChrimsonR-mCherry* was generated by replacing a NheI-AscI fragment containing EGFP of *pAAV-CAG-FLEX-EGFP* [36] with a NheI-AscI fragment containing *ChrimsonR-mCherry* PCR-amplified from *pAAV-TRE-ChrimsonR-mCherry*, which was acquired from Addgene (#92207, a gift from Dr. Alice Ting) [46]. As a negative control for the optogenetic study, we injected AAV-*EF1α-DIO-hM3Dq-mCherry*, which was generated with a plasmid *pAAV-EF1α-DIO-hM3Dq-mCherry* provided by Dr. Bryan Roth, University of North Carolina [78]. Recombinant AAV vectors (AAV2-rh10) were produced using a triple-transfection, helper-free method and purified as described previously [22]. The titers of recombinant AAV vectors were determined by quantitative PCR: AAV-*CAG-DIO-jGCaMP7s*, $3.4 \times 10^{13}$; AAV-*TRE-jGCaMP7s*, $6.3 \times 10^{11}$; AAV-*TRE-EGFP*, $5.7 \times 10^{11}$; AAV-*U6-gAvp-EF1α-DIO-mCherry*, $6.4 \times 10^{12}$; AAV-*U6-gControl-EF1α-DIO-mCherry*, $2.6 \times 10^{12}$; AAV-*EF1α-DIO-synaptophysin::GFP*, $1.2 \times 10^{13}$; AAV-*CAG-FLEX-ChrimsonR-mCherry*, $1.5 \times 10^{13}$; and AAV-*EF1α-DIO-hM3Dq-mCherry*, $4.5 \times 10^{12}$ genome copies/ml. Stereotaxic injection of AAV vectors was performed as described previously [22]. Two weeks after surgery, we began monitoring the mice for their locomotor activity.

## In vivo fiber photometry

We used 6 *Avp-CK1δ$^{-/-}$* × *Vip-tTA* (*Avp-Cre; CK1δ$^{flox/flox}$; Vip-tTA*) mice, 4 *Avp-CK1δ$^{-/-}$* (*Avp-Cre; CK1δ$^{flox/flox}$*) mice, and 11 control mice (8 *Avp-Cre; CK1δ$^{wt/flox}$; Vip-tTA* and 3 *Avp-Cre; CK1δ$^{wt/flox}$*). We combined the data of *Avp-Cre; CK1δ$^{flox/flox}$; Vip-tTA* and *Avp-Cre;*

$CK1\delta^{flox/flox}$ for AVP neurons recording considering the lack of intergroup differences. Additional 4 mice (*Vip-tTA*) were used for control experiments to measure EGFP signal in VIP neurons. The mice were anesthetized by administering a cocktail of medetomidine (0.3 mg/kg), midazolam (4 mg/kg), and butorphanol (5 mg/kg) and were secured at the stereotaxic apparatus (Muromachi Kikai). Lidocaine (8%) was applied for local anesthesia before making the surgical incision. We drilled small hole in the exposed region of the skull using a dental drill. We injected 0.5 to 1.0 μL of the virus (AAV-*CAG-DIO-jGCaMP7s* or AAV-*TRE-jGCaMP7s*) (flow rate = 0.1 μL/min) at the right SCN (posterior: 0.5 mm, lateral: 0.25 mm, depth: 5.7 mm from the bregma) with a 33 G Hamilton Syringe (1701RN Neuros Syringe, Hamilton) to label AVP neurons. Subsequently, we placed an implantable optical fiber (400 μm core, N.A. 0.39, 6 mm, ferrule 2.5 mm, FT400EMT-CANNULA, Thorlabs) above the SCN (posterior: 0.2 mm, lateral: 0.2 mm, depth: 5.2 to 5.4 mm from the bregma) with dental cement (Super-bond C&B, Sun Medical). The dental cement was painted black. Atipamezole (0.3 mg/kg) was administered postoperatively to reduce the anesthetized period. The mice were used for experiments 2 to 7 weeks after the virus injection and optical fiber implantation. Their ages were 3 to 10 months old, including both male and female.

A fiber photometry system (COME2-FTR, Lucir) was used to record the calcium signal of SCN neurons in freely moving mice [36,79]. Fiber-Coupled LED (M470F3, Thorlabs) with LED Driver (LEDD1B, Thorlabs) was used as an excitation blue light source. The light was reflected by a dichroic mirror (495 nm), went through an excitation bandpass filter (472/30 nm), then to the animal via a custom-made patch cord (400 um core, N.A. 0.39, ferrule 2.5 mm, length 50 cm, COME2-FTR/MF-F400, Lucir) and the implanted optical fiber. We detected the jGCaMP7s fluorescence signal by a photomultiplier through the same optical fibers and an emission bandpass filter (520/36 nm); furthermore, we recorded the signal using Power Lab (AD Instruments) with Lab Chart 8 software (AD Instruments). The excitation blue light intensity was 2.5 to 100 μW at the tip of the patch cord of the animal side. We recorded the same for 30 s every 10 min in 2 weeks to reduce photobleaching. During the recording, the mouse was housed in a 12-h light–dark cycle for more than 5 days (LD condition) and then moved to continuous darkness for approximately 10 days (DD condition) in a custom-made acrylic cage surrounded by a sound-attenuating chamber. A rotary joint for the patch cord was stopped during the recording to prevent artificial baseline fluctuation. The animal's locomotor activity was monitored using an infrared sensor (Supermex PAT.P and CompACT AMS Ver. 3, Muromachi Kikai).

The detected jGCaMP7s signal was averaged within a 30-s session [36]. To detrend the gradual decrease of the signal during recording days, ±12 h average from the time (145 points) was calculated as baseline (F). The data were subsequently detrended by the subtraction of F (ΔF). Then, the ΔF/F value was calculated. To determine the peak phase of jGCaMP7s calcium signal, ΔF/F were smoothened with a 21-point moving average, then the middle of the time points crossing value 0 upward and downward were defined as peak phases. Additionally, the intervals between peak phases were defined as the periods [22]. A double-plotted actogram of jGCaMP7s or EGFP signal was designed by converting all ΔF to positive values by subtracting the minimum value of ΔF. Subsequently, these values were multiplied with 100 or 1,000 and rounded off. The plots were made via ClockLab (Actimetrics) with normalization in each row. A double-plotted actogram of locomotor activity was also prepared and overlaid on that of jGCaMP7s signal.

The onset and offset of locomotor activity were determined using the actogram of locomotor activity. Initially, we attempted to automatically detect the onset and offset; however, it was followed by a manual visual inspection, along with modifications by the experimenter. To calculate CT of the peak phases of GCaMP signal, we have defined the regression line of locomotor activity onsets as CT12.

We confirmed the jGCaMP7s expression and the position of the optical fiber by slicing the brains in 100 μm coronal sections using a cryostat (Leica). The sections were mounted on glass slides with a mounting medium (VECTASHIELD HardSet with DAPI, H-1500, Vector Laboratories) and observed via epifluorescence microscope (KEYENCE, BZ-9000E).

### In vivo optogenetic stimulation with fiber photometry

We used 8 *Avp-Cre; Vip-tTA* mice ($n = 4$ for ChrimsonR, $n = 4$ for control, both male and female). We injected 1.0 μL of the mixture of viruses (AAV-*TRE-jGCaMP7s* with AAV-*CAG-Flex-ChrimsonR-mCherry* or AAV-*EF1α-DIO-hM3Dq-mCherry* and AAV-*TRE-jGCaMP7s*) at the right SCN (posterior: 0.5 mm, lateral: 0.25 mm, depth: 5.7 mm from the bregma) and then implanted an optical fiber (400 μm core, N.A. 0.39, 6 mm, Thorlabs) above the SCN (posterior: 0.2 mm, lateral: 0.2 mm, depth: 5.3 mm from the bregma) with dental cement. The mice were used for experiments more than 2 weeks after the surgery.

A fiber photometry system (FP3002, Neurophotometrics) was used to record the calcium signal of SCN neurons with optogenetic stimulation in freely moving mice [80,81]. Excitation light sources were a 470-nm LED for detecting calcium-dependent jGCaMP7s fluorescence signal (F470) and a 415-nm LED for calcium-independent isosbestic fluorescence signal (F415). The duration of excitation lights is 50 ms, and the onsets of the excitation timing of LEDs were interleaved. The lights passed through excitation bandpass filters, dichroic mirrors, and then to the animal via fiber-optic patch cords (BBP(4)_400/440/900-0.37_1m_FCM-4xFCM_LAF, MFP_400/440/LWMJ-0.37_1m_FCM-ZF2.5_LAF, Doric Lenses) and the implanted optical fiber. Subsequently, both signals were detected using a CMOS camera through the optical fibers, dichroic mirrors, and emission bandpass filters. The recorded signals were acquired using Bonsai software, with a sampling rate of 10 Hz for each color. The excitation intensities of the 470-nm and 415-nm LED at the animal side's patch cord tip were 130 μW and 90μW, respectively. Additionally, a 635-nm red laser (inside the fiber photometry system FP3002) was transmitted through the same optical fibers with an intensity of 2 mW. During the 720-s recording at ZT22, optical stimulation (635 nm, 50 ms pulse, 5 Hz, 120 s, 600 pulse count) was applied in the middle of the recording. Throughout the experiment, the mice were housed in a custom-made acrylic cage surrounded by a sound-attenuating chamber and maintained in a 12-h light–dark (LD) cycle.

The recorded data were interleaved to eliminate artifacts caused by red laser stimulation, and half of it was discarded. Consequently, the final sampling rate for the jGCaMP7s fluorescence signals at the 470-nm light excitation (F470) and the 415-nm excitation (F415) was 5 Hz. Ratio (R) was defined as the ratio between F470 and F415 (F470/F415) for calibration and reducing motion artifacts. The baseline ratio ($R_0$) was calculated as the mean R-value during the pre-laser stimulation period (−30 s to 0 s from the start of the laser stimulation, 150 points). $R_1$ was calculated as the mean R-value during the late period of laser stimulation (90 s to 120 s from the start of the laser stimulation, 150 points). The response $\Delta R/R_0\%$ was computed as the difference between $R_1$ and $R_0$ ($\Delta R$) divided by $R_0$ ($\Delta R / R_0$). All data analysis was done using MATLAB. After the recordings were completed, we histologically confirmed the jGCaMP7s and ChrimsonR-mCherry expressions and the position of the optical fiber.

### Immunohistochemistry

We used 15 *Avp-CK1δ$^{-/-}$; Rosa26-LSL-Cas9-2A-EGFP* mice for the *Avp* knockdown study (S5 Fig, 9 or 6 for g*Avp* or g*Control*) and 2 *Avp-Cre* mice for the Synaptophysin::GFP tracing study (Fig 7A and 7B). Immunostaining was performed as described previously [22,40,42]. Mice were killed approximately at ZT8 by transcardial perfusion of PBS followed by 4%

paraformaldehyde (PFA) in PBS. Then, the mice brains were postfixed in the 4% PFA at 4°C overnight, followed by immersion in 30% sucrose solution at 4°C for 2 days. Serial coronal brain sections (30 mm thickness) were made with a cryostat (CM1860, Leica) and collected in 4 series—one of which was further immunostained. For immunofluorescence staining, sections were washed with PBS containing 0.3% Triton X-100 (PBST) and blocked with PBST plus 3% BSA (blocking solution). Then, slices were incubated overnight with the designated primary antibodies in the blocking solution at 4°C. Antibodies used were rabbit anti-AVP antibody (1:4,000; AB1565, Merck Millipore) and rabbit anti-VIP antibody (1:1,000; #20077, Immunostar). Then, slices were washed with PBST, followed by incubation with the designated secondary antibodies in blocking solution for 4 h. Secondary antibodies used in this study were Alexa Fluor 488–conjugated donkey anti-rabbit IgG antibody (1:2,000, A-21206; Thermo Fisher Scientific) and Alexa Fluor 594–conjugated donkey anti-rabbit IgG antibody (1:2,000, A-21207; Thermo Fisher Scientific). After incubation with secondary antibody, slices were washed with PBS, mounted on slide glasses, air dried, and coverslipped using Mounting Medium (H-1500, Vector Laboratories; Dako Fluorescence Mounting Medium, Agilent Technologies). Images were taken using an epifluorescence microscope (BZ-9000, Keyence) or a confocal microscope (Fluoview Fv10i, Olympus). The AVP expression levels were quantified by Photoshop (Adobe) as follows [40]. First, the images were transformed to grayscale. Then, the mean intensities of pixels within the SCN were calculated. Finally, the values of the region lateral to the SCN were regarded as background and were subtracted from those of the SCN. When slices from *Avp-CK1δ$^{-/-}$; Rosa26-LSL-Cas9-2A-EGFP* mice were immunostained for AVP in green, green fluorescence was emitted from both AVP immunostaining and EGFP. Therefore, to evaluate the AVP expression level, EGFP fluorescence was quantified in negative control slices, i.e., mock-treated slices without incubation with anti-AVP antibody, and compared to the fluorescent intensity of slices incubated with anti-AVP antibody.

## Statistical analysis

All results are expressed as mean ± SEM. For comparisons of 2 groups, two-tailed Student or Welch *t* tests or Mann–Whitney *U* tests were performed. For comparisons of multiple groups with no difference of variance by Bartlett test, three-way or two-way repeated measures ANOVA or one-way ANOVA followed by post hoc Ryan test were performed. For comparisons of multiple groups with difference of variance by Bartlett test, nonparametric tests, Kruskal–Wallis test with post hoc Mann–Whitney *U* test with Bonferroni correction, Friedman rank sum test followed by Wilcoxon signed rank test with Bonferroni correction, and Wilcoxon signed rank test were performed with EZR, a graphical user interface for R [82] or ANOVA4. For circular data, Rayleigh test, Watson–Williams test, and Harrison–Kanji test were performed with Circstat MATLAB Toolbox for Circular Statistics [83]. All *P* values less than 0.05 were considered as statistically significant. Only relevant information from the statistical analysis was indicated in the text and figures.

## Supporting information

**S1 Fig. *CaMKIIα-Bmal1$^{-/-}$* mice are arrhythmic in DD, related to Fig 1.** (**A**) Representative locomotor activity of control and *CaMKIIα-Bmal1$^{-/-}$* mice (home-cage activity). Gray shading indicates the time when lights were off. (**B**) Representative periodograms of the locomotor activity rhythms of control (left) and *CaMKIIα-Bmal1$^{-/-}$* mice (right) in DD.
(TIF)

**S2 Fig. Reevaluation of circadian behavior rhythms in *Avp-CK1δ*<sup>−/−</sup> and *Avp-Bmal1*<sup>−/−</sup> mice by wheel-running activity, related to Figs 1 and 2.** (**A**) Representative wheel-running activity of control and *Avp-CK1δ*<sup>−/−</sup> mice. Gray shading indicates the time when lights were off. (**B**) The free-running period of wheel-running activity in DD. Values are mean ± SEM; $n = 4$ for control, $n = 5$ for *Avp-CK1δ*<sup>−/−</sup> mice. ***$P < 0.001$ by two-tailed Student *t* test. (**C**) Left: Actograms of the wheel-running activity of 3 control and 8 *Avp-Bmal1*<sup>−/−</sup> mice (M, male; F, female). Gray shading indicates the time when lights were off. Right: Periodograms of the individual wheel-running activity rhythms in the last 7 days in DD (vertical black lines). Most *Avp-Bmal1*<sup>−/−</sup> mice were arrhythmic in the last part of recording in DD.
(TIF)

**S3 Fig. Single-pixel trajectories of Per2::LUC oscillation in SCN slices of *CK1δ* conditional knockout mice, related to Fig 3.** Representative bioluminescence data of 15 pixels in a row along the mediolateral axis within ROIs (15 × 15 pixels) are shown for each region and mouse line. All data were detrended, smoothed, and aligned at ZT12 on the day of slicing as starting points. Black vertical lines labeled day 1 indicate projected ZT0. These data were used for statistical analysis in Fig 3.
(TIF)

**S4 Fig. Individual trajectories of GCaMP and EGFP fluorescence in SCN neurons in vivo, related to Figs 4, 5 and 6.** (**A**, **B**) Continuous recordings of GCaMP fluorescence from SCN AVP neurons (**A**) or VIP neurons (**B**) for 15 days (5 days in LD, 10 days in DD). Red, *Avp-CK1δ*<sup>−/−</sup> ($n = 6, 4$); blue, control (*Avp-CK1δ*<sup>+/−</sup>, i.e., *Avp-Cre; CK1δ*<sup>wt/flox</sup>, $n = 5, 6$). (**C**) Continuous recordings of EGFP fluorescence from SCN VIP neurons for 5 days (2 days in LD, 3 days in DD). Gray, individual trace; black, average.
(TIF)

**S5 Fig. AVP in the SCN is dispensable for the lengthening of behavioral free-running period in *Avp-CK1δ*<sup>−/−</sup> mice, related to Fig 7.** (**A**) Schematic diagram of viral vector (AAV-*U6-gAVP-EF1α-DIO-mCherry* or AAV-*U6-gControl-EF1α-DIO-mCherry*) injection at SCN in *Avp-CK1δ*<sup>−/−</sup> (*Avp-Cre; CK1δ*<sup>.flox/flox</sup>*; Rosa26-LSL-Cas9-2A-EGFP*) mice. (**B**) Representative coronal slices with (*gAvp*) or without (*gControl*) *Avp* knockdown in the SCN stained with an anti-AVP antibody (green), showing reduced AVP immunoreactivity. Bottom slices (no antibody) are the coronal slices mock-stained without anti-AVP antibody to evaluate EGFP expression (green) derived from the *Rosa26-LSL-Cas9-2A-EGFP* allele. The white rectangles indicate the position of the enlarged images of (**D**-**F**). Scale bar, 1 mm. (**C**) Green fluorescence intensity of SCN slices immunostained (*gAvp* or *gControl*) or mock-stained (no antibody) for AVP. Note that the green fluorescence of immunostained slices is the sum of fluorescence derived from AVP immunoreactivity and EGFP. AVP expression is reduced to the background level in slices with *Avp* knockdown. $n = 9$ for *gAvp*, $n = 6$ for *gControl*, $n = 6$ for no antibody. ***$P < 0.001$ by one-way ANOVA with post hoc Ryan test. (**D**-**F**) Enlarged images of the SCN, supraoptic nucleus (SON), and the paraventricular nucleus of the hypothalamus (PVH) in *gAvp* (**D**), *gControl* (**E**), and no antibody (**F**) conditions. Scale bar, 200 μm. (**G**) Representative locomotor activity actograms of *Avp-CK1δ*<sup>−/−</sup> mice with (*gAvp*) or without (*gControl*) *Avp* knockdown in the SCN (home-cage activity). Before an AAV injection, mice were housed in LD for 1 w (preLD) and in DD for approximately 3 w (preDD). Subsequently, mice were turned back to LD condition and received a *gAvp* or *gControl* AAV injection surgery (arrows) and then were housed in LD for 2 w (postLD) and in DD for approximately 3 w (postDD). Gray shading indicates the time when lights were off. (**H**) Periods of locomotor activity in preDD (last 10 days) and postDD (last 10 days) with (*gAvp*) or without (*gControl*)

*Avp* knockdown in *Avp-CK1δ⁻ᐟ⁻* mice. Each color indicates different mice. *n* = 9 for *gAvp*, *n* = 6 for *gControl*. No significant difference by two-way repeated measures ANOVA. (**I**) No clear relationship between green fluorescence intensity and postDD period for *gAvp* (*n* = 9) and *gControl* (*n* = 6) animals.
(TIF)

**S1 Data. Numerical data.**
(XLSX)

## Acknowledgments

We thank D. R. Weaver for the *CK1δ^flox* mouse; G. Schütz for the *CaMKIIα-Cre* (*Camk2a*::*iCreBAC*) mouse; Z. J. Huang for the *Vip-ires-Cre* mouse; J. Takahashi for the *Per2*::*Luc* mouse; C. J. Weitz for the *Bmal1^flox* mouse; F. Zhang for the *Rosa26-LSL-Cas9-2A-EGFP* mice; H. Okamoto for technical support to generate *Avp-Cre* mouse; Penn Vector Core for *pAAV2-rh10*; D. Kim and GENIE Project for *pGP-AAV-CAG-FLEX-jGCaMP7s-WPRE*; H. Kwon for *pAAV-TRE-EGFP*; *pAAV-EF1α-DIO-synaptophysin*::*GFP* for R. D. Palmiter; *pAAV-TRE-ChrimsonR-mCherry* for A. Ting; *pAAV-EF1α-DIO-hM3Dq-mCherry* for B. Roth; and *pX333* for A. Ventura. We thank all lab members, including M. Fukushi, M. Kawabata, M. T. Islam, and Y. Nishiwaki.

## Author Contributions

**Conceptualization:** Yusuke Tsuno, Michihiro Mieda.

**Funding acquisition:** Yusuke Tsuno, Ayako Matsui, Michihiro Mieda.

**Investigation:** Yusuke Tsuno, Yubo Peng, Mohan Wang, Ayako Matsui, Mizuki Sugiyama, Takahiro J. Nakamura, Takashi Maejima, Michihiro Mieda.

**Resources:** Shin-ichi Horike, Kanato Yamagata, Takiko Daikoku.

**Supervision:** Michihiro Mieda.

**Writing – original draft:** Yusuke Tsuno, Michihiro Mieda.

**Writing – review & editing:** Michihiro Mieda.

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
