## [Editor Report · Decision Letter 0]

29 Nov 2022

Dear Dr Mieda, 

Thank you for submitting your manuscript entitled "In vivo recording of the suprachiasmatic nucleus dynamics reveals a dominant role of arginine vasopressin neurons in the circadian pacesetting" for consideration as a Research Article by PLOS Biology.

Your manuscript has now been evaluated by the PLOS Biology editorial staff as well as by an academic editor with relevant expertise and I am writing to let you know that we would like to send your submission out for external peer review.

Once your full submission is complete, your paper will undergo a series of checks in preparation for peer review. After your manuscript has passed the checks it will be sent out for review. To provide the metadata for your submission, please Login to Editorial Manager (https://www.editorialmanager.com/pbiology) within two working days, i.e. by Dec 01 2022 11:59PM.

Kind regards,

Lucas

Lucas Smith, Ph.D.

Associate Editor

PLOS Biology

lsmith@plos.org

---

## [Decision Letter · Decision Letter 1]

25 Jan 2023

Dear Dr Mieda,

Thank you for your patience while your manuscript "In vivo recording of the suprachiasmatic nucleus dynamics reveals a dominant role of arginine vasopressin neurons in the circadian pacesetting" was peer-reviewed at PLOS Biology. Your manuscript has been evaluated by the PLOS Biology editors, an Academic Editor with relevant expertise, and by several independent reviewers.

As you will see in the reviewer reports, which can be found at the end of this email, although the reviewers find the work potentially interesting and appreciate the range of approaches used here, they have also raised a number of important concerns that would need to be thoroughly addressed before we can consider your manuscript further for publication at PLOS Biology. We think that, in order to achieve the standards we are trying to achieve, a substantial amount of work, including the generation of new data, would be required, to strengthen and clarify the conclusions of the study. We think it would also be important to more carefully engage with previous literature on this topic and discuss how your findings fit in and add to this field in greater detail.

Given the extent of revision that would be needed, we cannot make a decision about publication until we have seen the revised manuscript and your response to the reviewers' comments. Your revised manuscript would need to be seen by the reviewers again, and we would be looking for their enthusiastic support for your study in light of any changes made. Please note that we would not engage the reviewers unless their main concerns have been addressed.

We expect to receive your revised manuscript within 3 months, however do let us know if you need an extension of that time to perform the necessary revisions. Alternatively, given the scale of the requested work, we would also understand if you prefer to pursue faster publication of this work elsewhere. Please email us (plosbiology@plos.org) if you have any questions or concerns, or would like to request an extension. 

**IMPORTANT - SUBMITTING YOUR REVISION**

*Re-submission Checklist*

*Published Peer Review*

*PLOS Data Policy*

*Blot and Gel Data Policy*

Sincerely,

Lucas

Lucas Smith, Ph.D.

Associate Editor

PLOS Biology

lsmith@plos.org

REVIEWS:

Reviewer #1: In this ms, the authors mainly attempted to explain the dominant role of AVP neurons in the regulation of the period of ensemble SCN by deleting CK1δ specifically in AVP, CaMKIIα, and VIP neurons in mice. The rhythm of locomotor activity and ex vivo PER2::LUC signals in the shell and core regions of SCN were recorded and analyzed. The rhythms of AVP and VIP neuronal [Ca2+]i are also recorded in AVP-CK1δ KO mice. Although these data demonstrate that AVP neurons play a vital role in regulating the period of cellular circadian rhythm in vivo, more data are needed to demonstrate how AVP neurons regulate rhythm of SCN.

Major concerns:

1 In this paper, the authors emphasized that AVP neurons regulate the period of cellular circadian rhythm of VIP neurons in vivo. However, further investigation of the relationships between AVP neurons and VIP neurons is needed. For example, the projection of AVP neurons to VIP neurons could be tested by retrograde or anterograde tracing. Also, it would be better to measure the [Ca2+]i rhythm of VIP neurons with activation or inhibition of AVP neurons.

2 It might be better to observe whether the period of behavioral rhythm is lengthened in AVP receptor knockout mice.

3 The author emphasizes that specific CK1δ deletion in AVP neurons is similar to CK1δ deletion in CaMKIIα neurons, so why not record AVP and VIP-neuronal [Ca2+]i rhythm in CaMKIIα-CK1δ−/− mice as positive control? And Vip-CK1δ−/− mice as negative control.

4 The [Ca2+]i rhythm of VIP neuron located in core SCN shows longer period in Avp-CK1δ−/− mice while the period of core SCN in Avp-CK1δ−/−; Per2::Luc mice is similar to control, how to explain the inconsistency?

5 According to the result of previously published in the Neuron (2015) and Current Biology (2016), AVP neurons played a vital role in the regulation of SCN rhythm, and deleting different genes result in different circadian phenotypes. Thus, why not explain the molecular mechanism of AVP neurons regulate circadian rhythm?

Minor concerns:

1 Fig.3 and Fig.4 may be integrated into one figure. 

2 In P23, a spelling error of "P <0.01" should be "P < 0.01"?

Reviewer #2: In this manuscript, the authors build on an earlier investigation in which they assessed how panSCN knockout of Ck1delta influence suprachiasmatic nuclei (SCN) and SCN-mediated behavioral rhythms in mice. They use standard approaches such wheel-running and ex vivo PER2:::LUC bioluminescence measures as well as in vivo examination of calcium rhythms in the SCN via fibre photometry. A range of genetic mouse models are employed and both male and female mice of these models are used. The strength of the manuscript is the range of approaches, but the outcomes are not clear. The authors have hypothesized as to why there are disparities in the results from different experimental settings, but they do not test these. 

1) The authors seem somewhat surprised that there are disparities between the period of behavioral rhythms and the period of the SCN (as determined ex vivo by measurement of rhythms in PER2::LUC). However, such differences have been documented for over 10 years (e.g. Hughes et al., PlosOne 2011) and indicate how difficult it can be to compare these two measures of circadian rhythms. In some instances, the periods are comparable, but in others (such as here), they are not. Clearly there are a lot of other factors are at play (such as does the knockout of CK1delta affect how well the SCN slice survives culturing, is the SCN reset by the culturing, etc which are not really considered by the authors). It is notable that to demonstrate the importance of VIP to SCN PER2::LUC rhythms, Maywood and colleagues had to leave the SCN culture to damp out over 10+ days. 

2) Examination of the behavioral rhythms indicates that the wheel-running records are not being very carefully scrutinized. In the supplemental Figure2, several examples of AVP-Bmal1-/- are shown. The author conclude that most of these mice are arrhythmic. However, a number of them give evidence of 'splitting' ie that two bouts of behavior coincident with lighs-on and lights-off are sustained for 5-10 days in constant dark (1, 2, 3, 7, and 8) and in one example (3), these merge and to give at least the impression of longer period (ie >24h) rhythms. Thus, the conclusion that they are by and large arrhythmic is a generalization and is an over simplification. This is important because the authors are relating behavioral activities to those of the SCN ex vivo. 

3) The studies detailed in this manuscript used male and female mice. Are these balanced in the various studies ie are there equal numbers of male and female used? This is important as the behavioral records in Figure S2 show clear sex differences (in the control animals, examples 2 and 3 are from female mice) and in the AVP-Bmal1-/- examples 4 and 5 are from female mice. Since wheel-running suppresses nocturnal rodent SCN activity (Meijer group) then the in vivo calcium measurements should reveal this sex differences in either controls or experimental animals. This needs to be analyzed and documented. 

Reviewer #3: This is a very elegant and nicely conducted study with high experimental rigor. The authors replicate prior work that deletion of CK1delta in AVP neurons of the SCN lengthens the behavioral rhythm period (home cage activity). Here, they extend this finding by adding additional comparisons of wheel running data to CK1delta deletion throughout the SCN or just in VIP neurons which did and did not lengthen behavior period. Analysis of PER2::LUC explant rhythms from these same genotypes revealed a period lengthening effect in the SCN shell but only in the first cycle, as previously reported. The most novel contribution is that of the heroic in vivo calcium imaging of the AVP-CK1delta KO mice in both AVP and VIP neurons with simultaneous measurement of behavioral rhythms. These data showed that both populations lengthen period in vivo in phase with the lengthened behavioral rhythm. The results support the idea that AVP neurons are important for determining the period length of SCN rhythms in vivo but that cellular molecular clocks of AVP neurons require input from other SCN neurons (or other brain regions, as in the in vivo case). Some of these ideas have been supported before in the literature (although not all are recognized or discussed in the manuscript). Additional strengths of the study include the high rigor in methodology and statistical analysis. However, the impact of the paper could be increased by the authors addressing the following concerns:

1. The issue of AVP/VIP core/shell signaling has been addressed in other studies, including findings that AVP signaling modulates SCN period and phase in a spatially specific manner (PMID: 29931690) and that AVP receptor antagonists reduce PER2::LUC amplitude (PMID: 27626074 and PMID: 32460660). Not only are these papers not discussed but there seems to be a missed opportunity for simple pharmacology experiments in these sophisticated genetic manipulations, especially given that much of the PER2::LUC results were known from the prior publication.

2. The Summary states that the rhythms of these mice "did not recapitulate the period lengthening …" which is a bit overstated given that the period of the first cycle of shell neurons was substantially lengthened (as previously published).

3. Throughout the paper there is this tendency to compare period effects among the three methods (two in vivo and one ex vivo; see direct statements on the bottom of page 13) - however, it is important to note that there are very important methodological differences in how period is calculated. For example, in the wheel running behavior, period is calculated on days 10-24 in DD. The explants however, come straight out of LD and period is really estimated one cycle at a time. So, this would be akin to measuring the activity onset to onset on the first day of DD, and so on. This period estimation has a similar problem for the calcium imaging which is also calculated differently (days 8-10 in DD). If the goal is to directly compare in vivo to ex vivo then maybe a missing comparison in the study is to measure GCaMP in the ex vivo condition (just like PER2::LUC) to then see if the results are similar. This would also address the issue of whether the seemingly disparate results are due to a focus on the TTFL output (PER2::LUC) versus a physiological output (calcium oscillations, indicative of neuronal activity), which can also be dissociated. In all, these are important things to consider before concluding that the "SCN slice data may not necessarily be relied upon to support arguments …"

---

## [Decision Letter · Decision Letter 2]

17 Jul 2023

Dear Dr Mieda,

Thank you for your patience while we considered your revised manuscript "In vivo recording of the suprachiasmatic nucleus dynamics reveals a dominant role of arginine vasopressin neurons in the circadian pacesetting" for publication as a Research Article at PLOS Biology. This revised version of your manuscript has been evaluated by the PLOS Biology editors, the Academic Editor and the original reviewers.

Overall we appreciate the efforts that have gone into this revision and think the manuscript has been strengthened in the process. The reviewers are also largely satisfied by the revision - however, I note that Reviewer 2 has one remaining comment that we think can be addressed with additional discussion. Based on the reviews, we are likely to accept this manuscript for publication, provided you satisfactorily address the remaining point raised by the Reviewer 2. Please also make sure to address the following data and other policy-related requests.

EDITORIAL REQUESTS

1) ETHICS STATEMENT: In the methods section of your manuscript, please update the ethics statement to include the specific national or international regulations/guidelines to which your animal care and use protocol adhered. Please note that institutional or accreditation organization guidelines (such as AAALAC) do not meet this requirement.

2) TITLE: We think your title could be edited to be slightly more streamlined, by deleting "the" before "suprachiasmatic" and "circadian". If you agree, we suggest you change the title to "In vivo recording of suprachiasmatic nucleus dynamics reveals a dominant role of arginine vasopressin neurons in circadian pacesetting"

We expect to receive your revised manuscript within two weeks. 

*Published Peer Review History*

*Press*

Sincerely,

Lucas

Lucas Smith, Ph.D.

Senior Editor,

lsmith@plos.org,

PLOS Biology

Reviewer remarks:

Reviewer #1: My concerns have been fully addressed.

Reviewer #2: The authors have responded well to my concerns and appear to have addressed points raised by the other reviewers. They have clarified how they have measured circadian rhythms in the different settings. Further they have stated more overtly the sample sizes for the different studies as well as the sex of mouse used. 

One small point I think they could comment further on is the complexity of the genetic background of the mice. Circadian can vary with mouse strain (ie mice without mutations or genetic alterations) and as the authors already allude to, these studies used complex crosses and backgrounds. It is unclear just how close to a C57Bl6 background the mice models used in this study are. Some of the experimental variability reported in this manuscript could arise from comparing mice that are now deviating away from a 'wild-type' background. 

Reviewer #3: 

Authors have done an outstanding job of responding to all reviewers and have added new and important data and analysis to the manuscript. Regarding Rev 3 comments, authors have show that losing AVP as a neuropeptidergic signal does not shorten the behavioral free-running period of Avp-CK1δ−/− mice (new Figure S5). In addition, they now analyze the day-by-day change of periods of the in vivo Ca rhythms in DD (new Figure 6H-I), showing stable lengthening of period in vivo. Authors have altered sentences in Summary, Results, and Discussion where comparisons are made between slice data and behavioral data.

added these arguments in the Discussion (Text-revision, line 346). We have also deleted the

---

## [Editor Report · Decision Letter 3]

28 Jul 2023

Dear Dr Mieda,

Thank you for the submission of your revised Research Article "In vivo recording of suprachiasmatic nucleus dynamics reveals a dominant role of arginine vasopressin neurons in circadian pacesetting" for publication in PLOS Biology. Your revised manuscript has now been assessed by the PLOS Biology editorial team and by the Academic Editor, and we are satisfied by the changes made in the most recent revision. Therefore, on behalf of my colleagues and the Academic Editor, Paul J Shaw, I am pleased to say that we can in principle accept your manuscript for publication, provided you address any remaining formatting and reporting issues. These will be detailed in an email you should receive within 2-3 business days from our colleagues in the journal operations team; no action is required from you until then. Please note that we will not be able to formally accept your manuscript and schedule it for publication until you have completed any requested changes.

PRESS

We frequently collaborate with press offices. If your institution or institutions have a press office, please notify them about your upcoming paper at this point, to enable them to help maximize its impact. If the press office is planning to promote your findings, we would be grateful if they could coordinate with biologypress@plos.org. If you have previously opted in to the early version process, we ask that you notify us immediately of any press plans so that we may opt out on your behalf.

Sincerely, 

Lucas Smith, Ph.D.

Senior Editor

PLOS Biology

lsmith@plos.org